# Intrinsic rewards explain context-sensitive valuation in reinforcement learning

Gaia Molinaro[1]*, Anne G. E. Collins[1,2]

**1** Department of Psychology, University of California, Berkeley, Berkeley, California, United States of America, **2** Helen Wills Neuroscience Institute, University of California, Berkeley, Berkeley, California, United States of America

* gaiamolinaro@berkeley.edu

## Abstract

When observing the outcome of a choice, people are sensitive to the choice's context, such that the experienced value of an option depends on the alternatives: getting $1 when the possibilities were 0 or 1 feels much better than when the possibilities were 1 or 10. Context-sensitive valuation has been documented within reinforcement learning (RL) tasks, in which values are learned from experience through trial and error. Range adaptation, wherein options are rescaled according to the range of values yielded by available options, has been proposed to account for this phenomenon. However, we propose that other mechanisms—reflecting a different theoretical viewpoint—may also explain this phenomenon. Specifically, we theorize that internally defined goals play a crucial role in shaping the subjective value attributed to any given option. Motivated by this theory, we develop a new "intrinsically enhanced" RL model, which combines extrinsically provided rewards with internally generated signals of goal achievement as a teaching signal. Across 7 different studies (including previously published data sets as well as a novel, preregistered experiment with replication and control studies), we show that the intrinsically enhanced model can explain context-sensitive valuation as well as, or better than, range adaptation. Our findings indicate a more prominent role of intrinsic, goal-dependent rewards than previously recognized within formal models of human RL. By integrating internally generated signals of reward, standard RL theories should better account for human behavior, including context-sensitive valuation and beyond.

## Introduction

When selecting among multiple alternatives, the subjective values of available options and their neural correlates tend to scale as a function of the range of other options in the choice set [1–6]. For example, imagine walking to an ice cream cart that normally sells 2 flavors: chocolate, your favorite, and vanilla. After excitedly ordering a scoop of chocolate, you discover that they are now also selling pistachio ice cream, which you like even more. Suddenly your satisfaction with your order drops, despite the fact that the objective value of the chocolate ice cream has not changed. This example illustrates the phenomenon that subjective valuation

**Data Availability Statement:** Data collected for the present article and codes to reproduce all results are available at https://osf.io/sfnc9/. Data from Gold et al. (2012) has been made available at https://osf.io/8zx2b/. Data from Bavard et al. (2021) and

Bavard et al. (2018) was made available by the authors of the original study at https://github.com/sophiebavard/Magnitude/ and https://github.com/hrl-team/range/ respectively.

**Funding:** In producing this work, GM was supported by a Regents Fellowship for Graduate Study from the University of California, Berkeley. AGEC was supported by grants from the National Institutes of Health (NIH-R01MH118279) and the National Science Foundation (NSF 2020844). The funders had no role in study design, data collection and analysis, decision to publish, or preparation of the manuscript.

**Competing interests:** The authors have declared that no competing interests exist.

depends on what other options are available, which we refer to here as context-sensitive valuation. First documented when option values are explicitly known, context-sensitive valuation is also evident when participants learn from experience through trial and error [7,8].

Context-sensitive valuation is argued to be adaptive. Given the limited size of the brain's neuronal population and the individual neurons' firing capacities, responding to stimuli in relative terms can improve the signal-to-noise ratio [9,10]. However, context-sensitive valuation can also result in irrational behavior when options are presented outside of their original context [7,11–17]. For example, if option $P$ probabilistically results in +1 or 0 points and option $N$ in 0 or −1 points, most rational theories predict that human subjects should select $P$ over $N$, no matter the probabilities. Nonetheless, humans reliably tend to select option $N$ over $P$ when the negative option $N$ was initially encountered as the best option in its context and the positive option $P$ was encountered as the worst option in its own context ([11,14]; see e.g., Figs 3A and 4A). An outstanding question, then, regards the computational mechanisms that result in these behavioral patterns.

Range adaptation has been proposed as the functional form of context-sensitive valuation. It assumes that options are rescaled according to the minimum and maximum option value in a given context [13]; this range may be learned over time [12] or acquired immediately for a given context [11]. Returning to the ice cream example, range adaptation would result in the rescaling of the value of chocolate according to the known minimum (your liking of the vanilla flavor) and maximum (your liking of pistachio)—resulting still in a positive, but blunted response to your order (Fig 1, top). Range adaptation enables more efficient learning within fixed contexts, but can lead to irrational preferences once outcomes are presented in novel contexts—as data from human participants consistently shows [11–13,17].

However, we argue that context-sensitive valuation could also be explained by a simpler heuristic mechanism: that reaching one's goal is intrinsically valuable, independently of external rewards. In the example above, this simple heuristic results in a similar effect to range adaptation, but through a separate cognitive process. If your goal walking to the ice cream cart is to get the best possible ice cream, the subjective reward after ordering chocolate when pistachio was available could be accounted for by a mixture of your goal-independent evaluation of chocolate and a sense of having failed your goal (Fig 1, top). A long-established construct in the realm of social and personality psychology [18], goals have recently attracted the attention of cognitive computational psychologists and neuroscientists, who have recognized their central role in defining the reward function in reinforcement learning (RL) [19–21]. Recent findings support this hypothesized role. In one experiment, McDougle and colleagues [22] showed that even abstract, one-shot encoded goals can be endowed with value and contribute to value-learning neural signals. Moreover, Juechems and colleagues [23] found that rewards were encoded in a goal-dependent manner in the anterior cingulate cortex while people were assessing whether to accept a task-relevant offer. Furthermore, the extent to which an outcome aligns with the current goal dominates the neural representation of value beyond absolute rewards [19,24]. These results validate the general notion that, in humans, the computation of value can be flexibly shaped through cognitive control according to current goals [19,25]. These findings call for an integration of goals into outcome evaluations.

Here, we develop a computational model that represents a novel account of context-sensitive valuation based on the notion of goals as intrinsic rewards and their centrality to value estimation. Our "intrinsically enhanced" model assumes that participants weigh both the absolute outcome experienced ("extrinsic reward," a unidimensional scalar) and an internally generated signal of goal achievement (a binary "intrinsic reward," in the same scale as the extrinsic reward) when learning. We show that intrinsically enhanced RL can explain the results of 3 previously published data sets (totaling 935 participants) as well as range adaptation models.

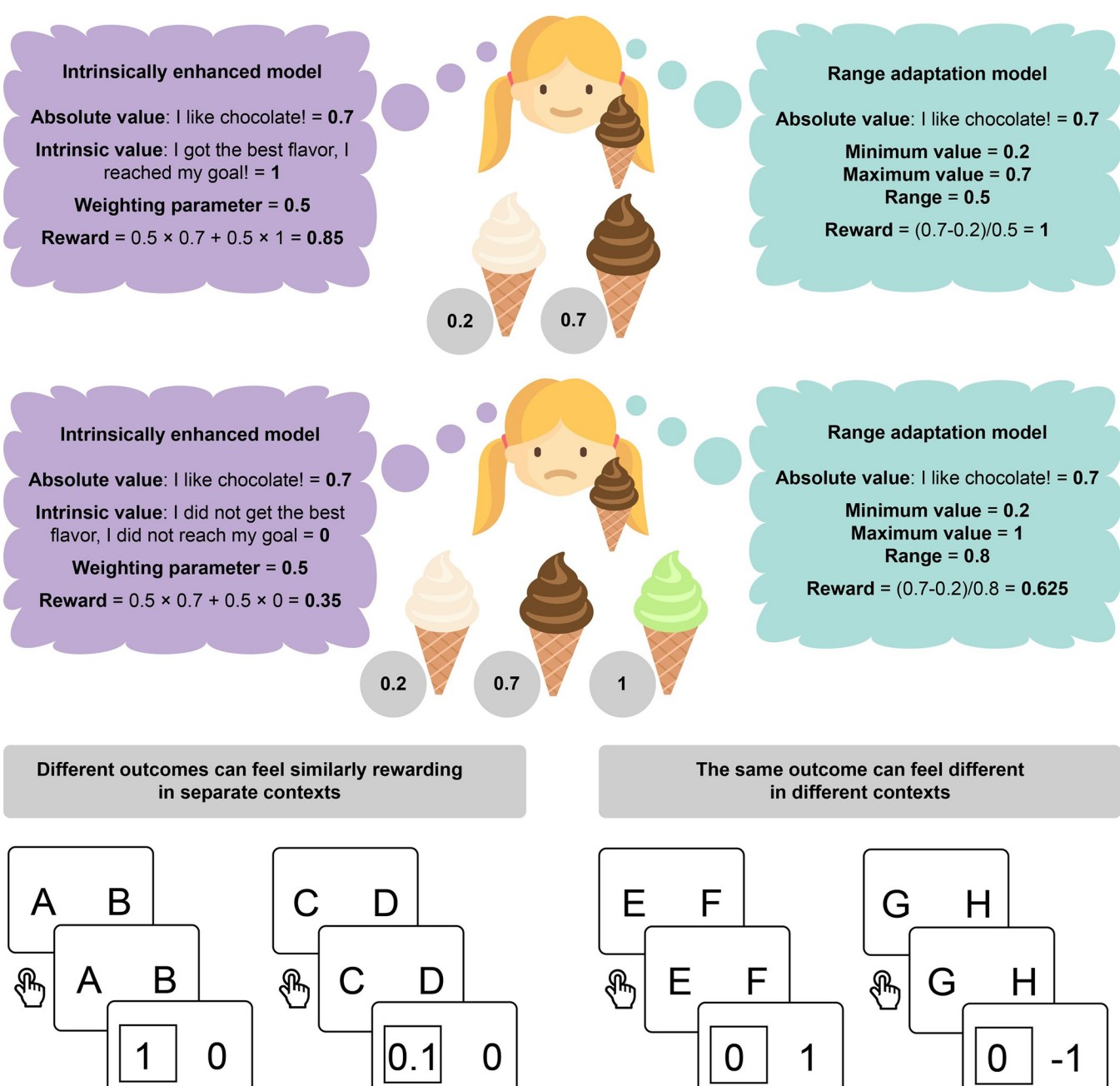

**Fig 1. Top**: The same outcome (getting chocolate ice cream) can lead to very different feelings of reward depending on the alternatives available at the time of choice. When chocolate is the best available option, it feels rewarding to get that flavor of ice cream, but when a better flavor (pistachio) is available, the feeling of reward for chocolate is dampened. This phenomenon may be explained through the intrinsic enhancement of absolute rewards based on goal achievement or failure (purple) or through a range adaptation mechanism (teal). In this situation, the 2 models make similar predictions but capture different cognitive processes. **Bottom left**: In a RL task, different outcomes (1 and 0.1) may feel similarly rewarding within their contexts when compared to a baseline (0) despite having different numeric values. **Bottom right**: The same outcome (0) may feel different in different contexts, where it is compared to different outcomes (1 and −1). RL, reinforcement learning.

Moreover, we find that the intrinsically enhanced model behaves more consistently with real participants in a fourth experimental design, compared to range adaptation. Finally, we test distinctive predictions of the intrinsically motivated model in a novel, preregistered

experiment. Our results in the latter—which we fully replicate in an independent sample and a control experiment—show that the intrinsically enhanced model captures behavior better than range adaptation.

## Results

### Candidate computational mechanisms

The aim of this study is to compare the explanatory power of range adaptation models to a newly proposed "intrinsically enhanced" model of context-sensitive valuation. Both models are based on a standard RL architecture, where the feedback received on each trial is compared to the predicted outcome to update the estimated value of available options. On each trial, this model chooses one of the available options based on their estimated value according to the softmax function [26]. Over time, a basic RL that updates value estimates via the delta rule can learn the optimal policy [27]. In updating the estimated value of each option, a standard, unbiased RL model equates reward with the objective outcome ($r$) associated with a given option.

While this simple algorithm is generally quite powerful, it cannot explain context-sensitive valuation. To account for the effects observed in humans, [12] proposed a range adaptation model that rescales outcomes according to the range of values yielded by other options. This range-adapted reward ($rr$) is obtained by subtracting the minimum value of a context ($s$) from the experienced outcome ($r$) and dividing this term by the difference between the maximum and the minimum outcome of the given context ($r_{max}(s)$ and $r_{min}(s)$, respectively):

$$rr_t = \frac{r_t - r_{min}(s)}{r_{max}(s) - r_{min}(s)} \tag{1}$$

We propose a different kind of subjective outcome that accounts for the importance of goals. Specifically, we assume that subjective outcomes reflect a mixture of the extrinsically provided reward (i.e., the objective outcome associated with a certain option) and a binary ("all-or-nothing"), internally generated signal ($ir$) that informs the agent of whether it has chosen the best option it could (thereby reaching its "goal"). This is calculated by comparing the outcome of the chosen option to the outcome of other options in the same context (either by taking counterfactual outcomes into account or by retrieving them from memory). A mixture parameter ($\omega$) regulates the extent to which each component contributes to the overall intrinsically enhanced reward ($ier$):

$$ier_t = \omega \cdot ir_t + (1 - \omega) \cdot r_t \tag{2}$$

"Intrinsically enhanced" and range adaptation models make similar qualitative predictions in most experimental settings, but they capture vastly different theoretical interpretations of the underlying phenomenon. Adjudicating which model best fits human behavior is thus an important step for understanding how context dependencies emerge in reinforcement learning and decision-making. To do so, we use quantitative model comparison [28] and posterior predictive checks, qualitatively comparing how well models capture behavioral patterns in the data [29].

We used hierarchical Bayesian inference (HBI) [30] to fit and compare the 2 models of interest. We also compare to other competing models, such as a hybrid actor–critic model that successfully captured context-sensitive behavior in previous experiments [14] and simpler models, such as a "win-stay/lose-shift" policy [31]. We ensure in each experiment that the different models are identifiable (see S7 Fig). Details about model implementation for each candidate mechanism are available in the Materials and methods alongside information about model fitting and comparison.

**Table 1. Summary information for each of the data sets used for data analysis and/or modeling.**

|  | N | Context | Stimuli | Most frequent reward | Probabilities | EV |
|---|---|---|---|---|---|---|
| **B21** | 800 | 1, 2 | [A, B], [C, D] | [10, 0] | [0.75, 0.25] | [7.5, 2.5] |
|  |  | 3, 4 | [E, F], [G, H] | [1, 0] | [0.75, 0.25] | [0.75, 0.25] |
| **B18** | 60 | 1 | [A, B] | [1, 0] | [0.75, 0.25] | [0.75, 0.25] |
|  |  | 2 | [C, D] | [0.1, 0] | [0.75, 0.25] | [0.075, 0.025] |
|  |  | 3 | [E, F] | [0, −1] | [0.75, 0.25] | [0, −0.25] |
|  |  | 4 | [G, H] | [0, −0.1] | [0.75, 0.25] | [−0.025, −0.075] |
| **G12** | 75 | 1 | [A, B] | [1, 0] | [0.9, 0.1] | [0.9, 0.1] |
|  |  | 2 | [C, D] | [1, 0] | [0.8, 0.2] | [0.8, 0.2] |
|  |  | 3 | [E, F] | [0, −1] | [0.9, 0.1] | [0.9, 0.1] |
|  |  | 4 | [G, H] | [0, −1] | [0.8, 0.2] | [0.8, 0.2] |
| **B22** | 50 | 1 | [A, B] | [14±2, 50±2] | [1, 1] | [14, 32] |
|  |  | 2 | [C, D, E] | [14±2, 32±2, 50±2] | [1, 1, 1] | [14, 32, 50] |
|  |  | 3 | [F, G] | [14±2, 86±2] | [1, 1] | [14, 86] |
|  |  | 4 | [H, I, J] | [14±2, 50±2, 86±2] | [1, 1, 1] | [14, 50, 86] |
| **M22** | 50 | 1 | [$L_1$, $M_1$, $H_1$] | [14±2, 50±2, 86±2] | [1, 1, 1] | [14, 50, 86] |
|  |  | 2 | [$L_2$, $M_2$, $H_2$] | [14±2, 50±2, 86±2] | [1, 1, 1] | [14, 50, 86] |

Previously collected data sets were originally reported by [12] (B21), [11] (B18), [14] (G12), and [13] (B22). For each data set experiment we used, the rewards and probabilities associated with each context and stimulus of the learning phase, as well as their EV are reported. The total number of participants in each original experiment (N) is also shown.

EV, expected value.

## Data sets and experimental designs

Seven data sets and/or experimental designs in total were used for analysis. Three were previously published [11,12,14], one was a task described in a preprint [13], one was an original study (M22), one was a replication of M22 (M22R), and one was a control version of M22 (M22B). Experimental designs differed in the exact task structure and reward contingencies, but all involved a learning phase, during which stimuli and their outcomes were presented within fixed contexts (e.g., the same pair or triplet of options), and a test phase in which options were presented in novel pairings. The key features of each data set (e.g., reward contingencies of each option) are summarized in Table 1. More detailed information about each data set and task design is reported in S1 Table.

## Data set B21: Bavard and colleagues [12]

**Task structure.** The data used for the first set of analyses was collected by Bavard and colleagues [12]. The experiment involved 8 variants of the same task, and each participant only completed a single variant. One hundred participants were recruited for each of the 8 variants of the task (800 in total). The task comprised a learning phase and a test phase (120 trials each). Within the learning phase, participants viewed 4 different pairs of abstract stimuli (contexts 1 to 4) and had to learn, for each pair and by trial and error, to select the stimulus that yielded a reward most often. Within each pair of stimuli, one yielded a reward 75% of the time, the other one 25% of the time. Possible outcomes were 0 or 10 in half of the pairs, and 0 or 1 in the rest of the pairs. Pairs of stimuli thus had differences in expected value (ΔEV) of either 5 or 0.5. During the test phase, stimuli were recombined so that items associated with maximum outcomes of 10 were pitted against items associated with maximum outcomes of 1, yielding pairs with ΔEVs of 6.75, 2.25, 7.25, and 1.75 (contexts 5 to 8). Each variant of the task was

characterized by whether feedback was provided during the test phase, the type of feedback displayed during the learning phase (if partial, feedback was only shown for the chosen option; if complete, counterfactual feedback was also displayed), and whether pairs of stimuli were presented in a blocked or interleaved fashion. The latter feature was consistent across phases of the same variant. The full task details are available at [12].

**Summary of the behavioral results.** Performance throughout the task was measured as the proportion of optimal choices, i.e., the proportion of trials in which the option with higher expected value (EV) in each pair was chosen. Participants' performance in the learning phase was significantly higher than chance (i.e., 0.5; M = 0.69 ± 0.16, t(799) = 32.49, $p < 0.001$), with a moderate effect of each pair's ΔEV such that participants performed better in pair with ΔEV = 5 (0.71 ± 0.18) than in pairs with ΔEV = 0.5 (0.67 ± 0.18, t(799) = 6.81, $p < 0.001$). Performance was also better than chance in the test phase (0.62 ± 0.17, t(799) = 20.29, $p < 0.001$), but this was highly dependent on each pair's ΔEV (F(2.84,2250.66) = 271.68, $p < 0.001$).

Context 8 (ΔEV = 1.75 pair) was of particular interest, as it illustrated apparently irrational behavior. Specifically, it paired a stimulus that was suboptimal to choose in the learning phase (25% 10 versus 0) with a stimulus that was optimal in the learning phase (75% 1 versus 0). However, the previously suboptimal stimulus became the better one in the testing phase (having EV = 2.5, compared to EV = 0.75).

If participants simply learned the absolute value of these stimuli, they should select the option with EV = 2.5 in the test phase. However, if participants learned the relative value of each option within the pair it was presented in, they should view the EV = 2.5 option as less valuable than the EV = 0.75 alternative. In support of the latter, participants' selection rate of the highest EV stimulus in test trials with ΔEV = 1.75 was significantly below chance (M = 0.42 ± 0.30, t(799) = −7.25, $p < 0.001$; "context 8"; Fig 2A).

**Both intrinsically enhanced and range adaptation mechanisms capture behavior well.** Simulating behavior with the intrinsically enhanced model replicated participants' behavioral signatures well (Fig 2A, purple), confirming its validity [28,29]. We sought to confirm that the intrinsic reward was instrumental in explaining the behavioral pattern. Indeed, in the intrinsically enhanced model, the ω parameter (M = 0.56 ± 0.01) was significantly correlated with signatures of context-sensitive learning, specifically, the error rate in context 8 (i.e., choosing a bandit with EV = 0.75 versus one with EV = 2.5; Spearman's ρ = 0.46, $p < 0.001$; S14A Fig). The range model also recovered behavior well in most experiments, but less accurately on average (Fig 2A, teal).

**Model comparison favors the intrinsically enhanced model.** To quantify the difference, we fit our models via HBI, which estimates individual parameters hierarchically while comparing candidate models. The model comparison step of HBI favored the intrinsically enhanced model as the most frequently expressed within the studied population, even after accounting for the possibility that differences in model responsibilities were due to chance (protected exceedance probability = 1). While each of the alternative models also provided the best fit in a fraction of the participants (model responsibilities: intrinsically enhanced = 0.34, range adaptation = 0.19, hybrid actor–critic = 0.20, unbiased RL model = 0.08, win-stay/lose-shift = 0.19), the responsibility attributed to the intrinsically enhanced model was highest (Fig 2B).

The intrinsically enhanced model further provides an explanation for why more successful learning would lead to less optimal behavior in the test phase. Both a blocked design and complete feedback would enable participants to more easily identify the context-dependent goal throughout learning, and thus rely less on the numeric feedback presented to them and more on the binary, intrinsic component of reward (i.e., whether the intended goal has been reached). Indeed, an ANOVA test revealed that the $\omega_{learn}$ parameter, which attributes relative importance to the intrinsic signal in the learning phase, was higher in experiments with a

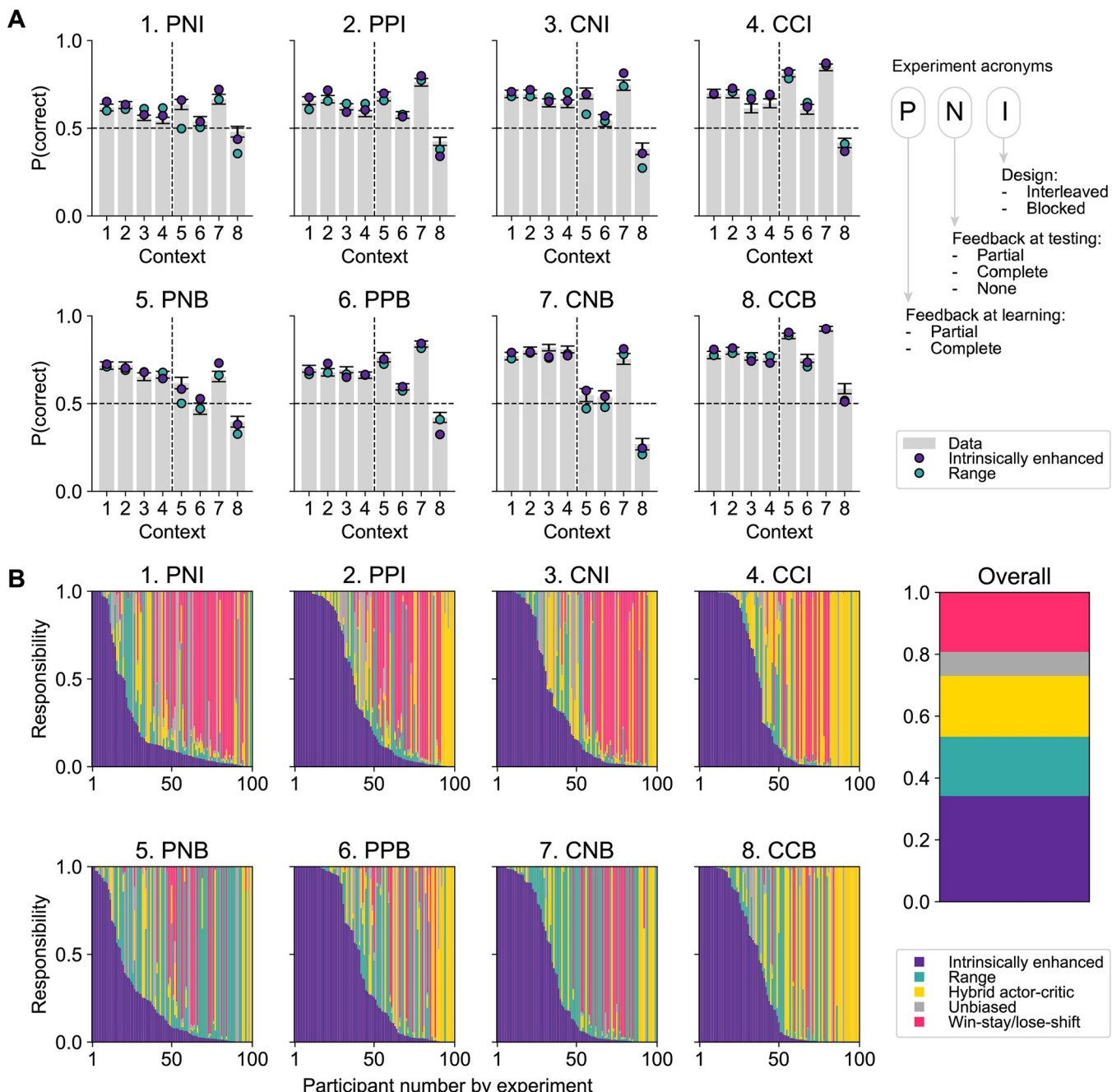

**Fig 2.** **(A)** Model validation by experimental condition and context with parameters extracted through Laplacian estimation [30], showing the simulated performance yielded by the intrinsically enhanced model (in purple) and the range adaptation model (in teal). The participants' data is shown in the gray bars. Contexts 1–4 refer to the learning phase and contexts 5–8 to the test phase. Overall, both the intrinsically enhanced model and the range adaptation model captured participants' behavior relatively well, the former outperforming the latter. Abbreviations: the first letter in each triplet indicates whether feedback was partial (P) or complete (C) during learning; the second letter indicates whether feedback in the test phase was partial (P), complete (C), or not provided (N); the third letter indicates whether the experimental design was interleaved (I) or blocked (B). Error bars indicate the SEM. **(B)** Model responsibilities overall and across experimental conditions. Data underlying this figure are available at https://github.com/hrl-team/range. Computational modeling scripts to produce the illustrated results are available at https://osf.io/sfnc9/.

blocked design (M = 0.59 ± 0.01) than in experiments where contexts were displayed in an interleaved fashion (M = 0.52 ± 0.01; t(799) = 3.62, $p < 0.001$; S8 Fig). At the same time, the $\omega_{learn}$ parameter was higher for participants who underwent the learning phase with complete feedback (M = 0.60 ± 0.01) than for those who only received partial feedback (M = 0.53 ± 0.01; t(799) = 4.07, $p < 0.001$). There was no significant interaction between the 2 factors (t(799) = −0.22, $p = 0.825$).

## Data set B18: Bavard and colleagues [11]

**Task structure.**   The B21 data set is well suited to study the rescaling of outcomes within a given context in the positive domain. To investigate whether the same behavior can be explained by either intrinsically enhanced or range adaptation models in the negative domain, we retrieved a data set reported in [11]. Here, subjects engaged with both gain- and loss-avoidance contexts. In gain contexts, the maximum possible reward was either 1 (for 1 pair of stimuli) or 0.1 (for another pair of stimuli). These rewards were yielded with 75% probability by selecting the better option in each pair, while the outcome was 0 for the remaining 25% of the time. Probabilities were reversed for selecting the worse option within a given pair. Loss avoidance trials were constructed similarly, except that the maximum available reward was 0 in both cases, while the minimum possible outcome was either −1 (in 1 pair) or −0.1 (in the other pair). This design effectively manipulates outcome magnitude and valence across the 4 pairs of stimuli presented during the learning phase. All possible combinations of stimuli were then presented, without feedback, in a subsequent testing phase. Data were collected on 2 variants of this experiment, which we analyze jointly. In Experiment 1 ($N = 20$), participants only received feedback on the option they selected. In Experiment 2 ($N = 40$), complete feedback (i.e., feedback on both the chosen and unchosen option) was presented to participants in 50% of the learning phase trials. Presentation order was interleaved for both experiments. The full task details are available at [11].

**Summary of the behavioral results.**   On average, participants selected the correct response more often than chance during the learning phase (t(59) = 16.6, $p < 0.001$), showing that they understood the task and successfully learned the appropriate stimulus-response associations in this context. Stimulus preferences in the test phase were quantified in terms of the choice rate for each stimulus, i.e., the average probability with which a stimulus was chosen in this phase. While stimuli with higher EV tended to be chosen more often than others (F(59) = 203.50, $p < 0.001$), participants also displayed irrational behaviors during the test phase. In particular, choice rates were higher for the optimal option of the 0.1/0 context (EV = 0.075; M = 0.71 ± 0.03) than choice rates for the suboptimal option of the 1/0 context (EV = 0.25; M = 0.41 ± 0.04; t(59) = 6.43, $p < 0.001$). Choice rates for the optimal option in the 0/−0.1 context (EV = −0.025; M = 0.42 ± 0.03) were higher than choice rates for the suboptimal option of the 0.1/0 context (EV = 0.025; M = 0.56 ± 0.03; t(59) = 2.88, $p < 0.006$). These effects show that, when learning about the value of an option, people do not simply acquire an estimate of its running average, but rather adapt it, at least partially, based on the alternatives the option is presented with.

**The intrinsically enhanced model captures behavior better than the range adaptation model.**   Both the intrinsically enhanced model and the range adaptation model captured the key behavioral patterns displayed by participants in the test phase, with the intrinsically enhanced model more closely matching their behavior than the range model (Fig 3A). The intrinsically enhanced model's $\omega$ parameter (M = 0.55 ± 0.03) was significantly correlated with key signatures of context-sensitive learning (i.e., the average error rate when choosing between a bandit with EV = 0.075 versus one with EV = 0.25 and between a bandit with EV = −0.025

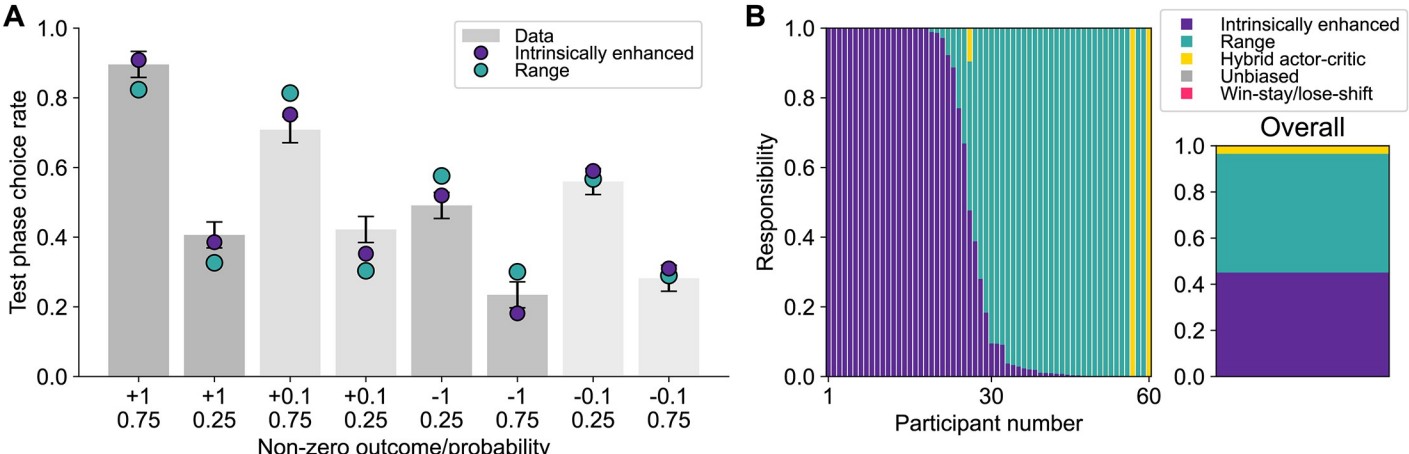

**Fig 3.** **(A)** Model validation by context with parameters extracted through Laplacian estimation, showing the simulated performance yielded by the intrinsically enhanced model (in purple) and the range adaptation model (in teal), overlaid with the data (gray bars). The intrinsically enhanced model outperformed the range model in capturing participants' behavior in the test phase, although both expressed the key behavioral pattern displayed by participants. Bars indicate the SEM. **(B)** Model responsibilities across participants. Data underlying this figure are available at https://github.com/sophiebavard/Magnitude/. Computational modeling scripts to produce the illustrated results are available at https://osf.io/sfnc9/.

versus one with EV = 0.025; Spearman's $\rho$ = 0.51, $p < 0.001$; S14B Fig), confirming the role of intrinsic reward in explaining context-sensitivity effects. We note that, consistent with our findings, the winning model in [11] was a hybrid between relative and absolute outcome valuation which, in mathematical terms, was equivalent to the intrinsically enhanced model. However, as discussed below (see Discussion), the theory behind the hybrid model presented in [11] aligns more closely with the range adaptation mechanism, and the mathematical overlap only exists for data set B18.

**Model comparison slightly favors the range adaptation model.** Range adaptation was the most frequently expressed model within the studied population (protected exceedance probability = 0.70). Although the HBI pipeline has been shown to avoid overpenalizing complex models [30], it is still possible that the additional complexity of the intrinsically enhanced model was not sufficient to compensate for its better ability to capture behavior. Nonetheless, a large proportion of the participants was best fit by the intrinsically enhanced model (model responsibilities: intrinsically enhanced = 0.45, range adaptation = 0.52, hybrid actor–critic = 0.03, unbiased RL model = 0, win-stay/lose-shift = 0; Fig 3B).

### Data set G22: Gold and colleagues [14]

**Task structure.** Data sets B21 and B18 were collected by the same research group. To exclude the possibility that analyses on these data sets may be due to systematic features of the experimenters' design choices, we employed a data set collected separately by Gold and colleagues [14]. This data set comprised both healthy controls ($N = 28$) and clinically stable participants diagnosed with schizophrenia or schizoaffective disorder ($N = 47$). This allowed us to test whether our findings could be generalized beyond the healthy population. In this task, 4 contexts were presented to participants in an interleaved fashion. These contexts were the result of a $2 \times 2$ task design, where the valence of the best outcome and the probability of obtaining it by selecting the better option were manipulated. Across contexts, the best outcome in a given context was either positive ("Win!", coded as 1) or neutral ("Keep your money!", coded as 0), and the worst outcome was either neutral ("Not a winner. Try again!", coded as 0)

or negative ("Loss!", coded as −1), respectively. The better option yielded the favorable outcome either 90% or 80% of the time and yielded the unfavorable outcome either 10% or 20% of the time, respectively. Following a learning phase, in which options were presented only within their context and participants received partial feedback, all possible pairs of stimuli were presented in a testing phase to participants, who received no feedback upon selecting one of them. The full task details are available at [14].

**Summary of the behavioral results.** On average, participants selected the correct response more often than chance in the learning phase (i.e., 0.5; M = 0.73 ± 0.15; t(74) = 13.42, $p < 0.001$). In the test phase, participants selected the better option more often than predicted by chance (M = 0.65 ± 0.11; t(74) = 11.62, $p < 0.001$). However, they also displayed irrational behavior, in that performance was below 0.5 when the optimal options in the 0/−1 contexts (EV = −0.10, −0.20) were pitted against the suboptimal options in the 1/0 contexts (EV = 0.10, 0.20; M = 0.4 ± 18; t(74) = −4.69, $p < 0.001$). Once again, these effects illustrate an adaptation in the value of presented options based on the alternatives offered in the same context, as predicted by range adaptation and intrinsically enhanced, but not simple RL algorithms.

**Both the intrinsically enhanced model and the range adaptation model capture behavior adequately.** Both main candidate models adequately captured the participants' behavior in the test phase (Fig 4A). Again, the intrinsically enhanced model's ω parameter (M = 0.46 ± 0.01) was significantly correlated with signatures of context-sensitive learning (i.e., the average error rate when choosing between a bandit with EV = −0.1 versus one with EV = 0.1 and between a bandit with EV = −0.2 versus one with EV = 0.2; Spearman's $\rho = 0.53$, $\rho < 0.001$; S14C Fig).

**Model comparison slightly favors the intrinsically enhanced model.** The intrinsically enhanced model was the most frequently expressed within the studied population (protected exceedance probability = 0.84). Both the intrinsically enhanced and the range adaptation models had high responsibility across participants (intrinsically enhanced = 0.50, range adaptation = 0.39, hybrid actor–critic = 0, unbiased RL model = 0.01, win-stay/lose-shift = 0; Fig 4B).

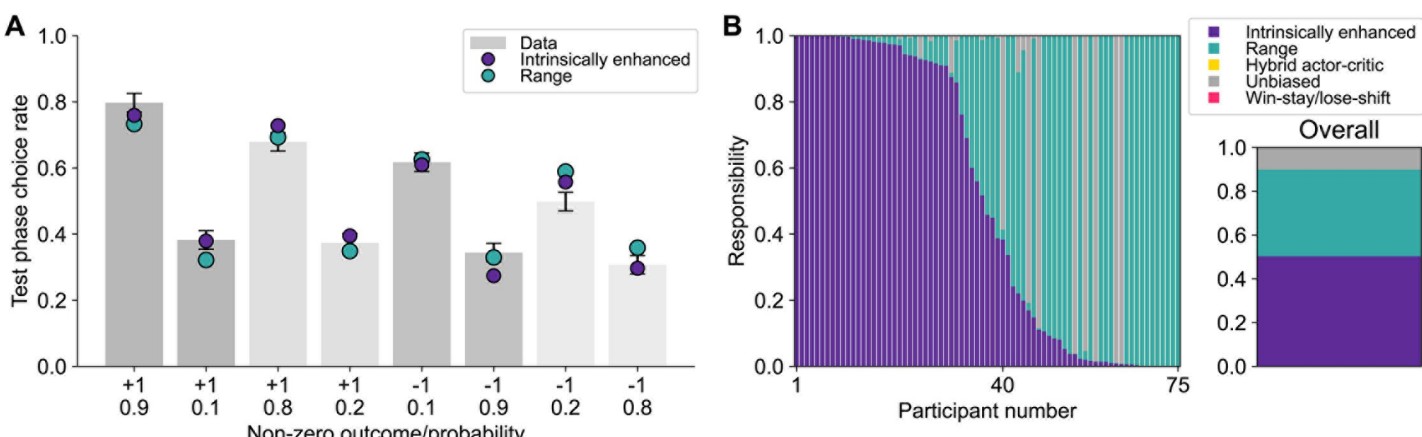

**Fig 4.** **(A)** Model validation by context with parameters extracted through Laplacian estimation, showing the simulated performance yielded by the intrinsically enhanced model (in purple) and the range adaptation model (in teal), overlaid with the data (gray bars). The intrinsically enhanced model outperformed the range model in capturing participants' behavior in the test phase, although both expressed the key behavioral pattern displayed by participants. Bars indicate the SEM. **(B)** Model responsibilities across participants. Data underlying this figure are available at https://osf.io/8zx2b/. Computational modeling scripts to produce the illustrated results are available at https://osf.io/sfnc9/.

## Data set B22: Bavard and Palminteri [13]

**Task structure.**   In all the data sets described above, stimuli were only ever presented in pairs. To test whether the models of interest might make different predictions where outcomes are presented in larger contexts, we used a task design presented by Bavard and Palminteri [13]. At the time of writing, this work was not published in a peer-reviewed journal, and data for this experiment was not publicly accessible. Therefore, we only provide ex ante simulations for this task. The study reported in the preprint involved 3 different variants of the same task, 2 of which included forced-choice trials. Since we did not have access to the precise sequence of stimuli participants viewed in experiments with forced trials, we focused on the first experiment reported by the authors. Here, participants ($N = 50$) were presented with contexts composed of either 2 (binary) or 3 stimuli (trinary), wherein each stimulus gave rewards selected from a Gaussian distribution with a fixed mean and a variance of 4. The range of mean values each stimulus could yield upon selecting it was either wide (14–86) or narrow (14–50). For trinary trials, the intermediate option value was either 50 or 32 in wide and narrow contexts, respectively. Participants first learned to select the best option in each context via complete feedback. They did so for 2 rounds of learning, and contexts were presented in an interleaved manner. The stimuli changed between learning sessions, requiring participants to re-learn stimulus-reward associations, but the distribution of outcomes remained the same across sessions. Then, they were presented with all possible pairwise combinations of stimuli from the second learning session. No feedback was displayed during this test phase. The full task details are available at [13].

**Summary of the behavioral results.**   The authors [13] report that on average, participants selected the correct response more often than chance (0.5 or 0.33, depending on whether the context was binary or trinary) in the learning phase. Overall performance in the test phase was also better than chance, showing that participants were able to generalize beyond learned comparisons successfully. Behavioral patterns showed signatures of both simple RL and range adaptation models. On the one hand, as predicted by a simple RL model, options with the highest value in narrow contexts (EV = 50) were chosen less often than options with the highest value in wide contexts (EV = 86). On the other hand, as predicted by the range adaptation model, the mid-value option in the wide trinary context (EV = 50) was chosen less frequently than the best option in the narrow binary context (EV = 50). Moreover, in trinary contexts, the choice rate of mid-value options was much closer to the choice rate of the lowest-valued options than would be expected by an unbiased RL model. To capture this effect, the authors introduced an augmented variant of the range adaptation model that incorporates a nonlinear transformation of normalized outcomes [13]. Below, we ask whether the intrinsically motivated model might capture all these effects more parsimoniously.

**Predictions from the intrinsically enhanced model capture participants' behavior.**   Fig 5 illustrates the predictions made by the unbiased, intrinsically enhanced, and range adaptation model (with an additional parameter, here called $z$, which captures potential nonlinearities in reward valuation in the range adaptation model). On the one hand, the unbiased RL model correctly predicts that choice rates for high-value options in narrow contexts (EV = 50) would be lower than for high-value options in wide contexts (EV = 86), while the range adaptation model does not. Moreover, the unbiased model correctly predicts that choice rates for the mid-value options would be higher in the wide (EV = 50) than in the narrow context (EV = 32), while the range adaptation model selects them equally often. On the other hand, the range adaptation model correctly predicts that participants' choice rates for mid-value options in the wide trinary context (EV = 50) would be lower than those for high-value options in the narrow binary context (EV = 50), while the unbiased RL model does not. With the addition of

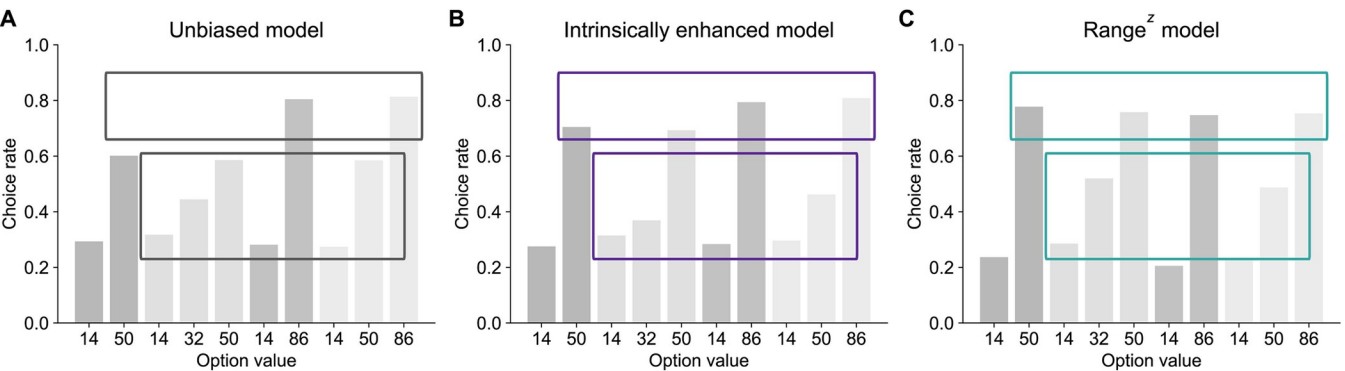

**Fig 5. Predictions made by the unbiased, intrinsically enhanced, and range$^z$ models. (A)** The unbiased model correctly predicts lower choice rates for high-value and mid-value options in wide than in narrow contexts (upper gray box), but incorrectly predicts similar choice rates for the option with value 50 regardless of context (lower gray box). **(B)** The intrinsically enhanced model captures all behavioral patterns found in participants' data [13]. It correctly predicts lower choice rates for high-value and mid-value options in wide than in narrow contexts (upper purple box) and correctly predicts higher choice rates for the option with value 50 in the trinary narrow context than in the trinary wide context (lower purple box). It also predicts that choice rates for mid-value options will be closer to those of low-value options than high-value options (lower purple box). **(C)** The range$^z$ model correctly predicts higher choice rates for the option with value 50 in the trinary narrow context than in the trinary wide context (lower teal box), but incorrectly predicts similar choice rates for high-value options in the narrow and wide trinary contexts (upper teal box). Simulation scripts used to produce this figure are available at https://osf.io/sfnc9/.

the nonlinearity parameter $z$, the range adaptation model can also capture the fact that choice rates for mid-value options in trinary contexts are closer to those of low-value options than those of high-value options, while the unbiased RL model cannot. Overall, [13] provide convincing evidence that range adaptation mechanisms surpass other classic and state-of-the-art models, including standard RL algorithms and divisive normalization, making it a strong model of human context-sensitive valuation. However, only the intrinsically enhanced model can capture all the key effects displayed by participants. Although these predictions await validation through fitting on the collected data, the intrinsically enhanced model succinctly explains the different behavioral signatures observed in real participants better than other candidate models.

## Data set M22: Novel task distinguishing between intrinsically enhanced and range adaptation models

**Task structure.** None of the data sets described above were collected to distinguish between intrinsically enhanced and range adaptation models. To qualitatively, as well as quantitatively disentangle the 2, we conducted an additional, preregistered experiment (henceforth, M22; the preregistered analysis pipeline and hypotheses are available at https://osf.io/2sczg/). The task design was adapted from [13] to distinguish between the range adaptation model and the intrinsically enhanced model (Fig 6). Participants ($N = 50$, plus a replication sample of 55 participants—see S1 Text) were tasked with learning to choose the optimal symbols out of 2 sets of 3 (2 trinary contexts). On each trial of the learning phase, they chose 1 stimulus among either a pair or a group of 3 stimuli belonging to the same context and received complete feedback. As in B22, each stimulus was associated with outcomes drawn from a Gaussian with fixed means and a variance of 4. Within each context was a low-value option (L) with a mean of 14, a middle-value option (M) with a mean of 50, and a high-value option (H) with a mean of 86. Thus, there was a pair of equivalent options across contexts. Both sets of 3 (i.e., stimuli $L_1$, $M_1$, and $H_1$ for context 1, and $L_2$, $M_2$, and $H_2$ for context 2) were presented 20 times each. However, the number of times each pair of stimuli was presented differed among contexts. Specifically, the $M_1$ stimulus was pitted against the $L_1$ stimulus 20 times, and never against the

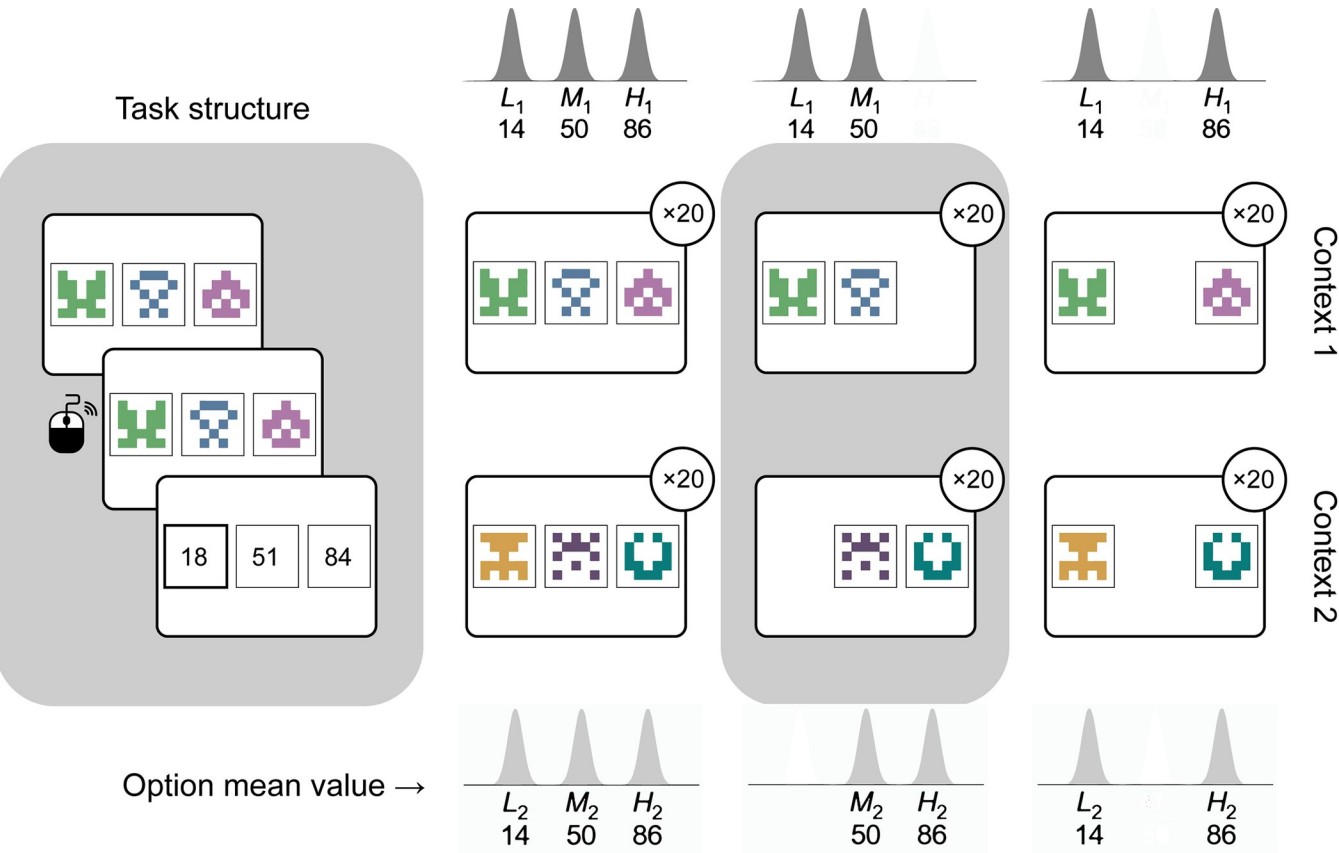

**Fig 6. Left**: Task structure. Participants viewed available options, indicated their choice with a mouse click, and viewed each option's outcome, including their chosen one highlighted. **Right**: Experimental design. Both context 1 (top row) and context 2 (bottom row) contained 3 options, each having a mean value of 14, 50, or 86. The contexts differed in the frequency with which different combinations of within-context stimuli were presented during the learning phase (gray shaded area). In particular, while option $M_1$ (EV = 50) was presented 20 times with option $L_1$ (EV = 14), option $M_2$ (EV = 50) was presented 20 times with option $H_2$ (EV = 86). Intuitively, this made $M_1$ a more frequent intrinsically rewarding outcome than $M_2$. The 2 contexts were otherwise matched.

$H_1$ stimulus, while the $M_2$ stimulus was pitted against the $H_2$ stimulus 20 times, and never against the $L_2$ stimulus. Both L options were pitted against the respective H option 20 times (see Fig 6). All possible pairs of stimuli were presented 4 times in the test phase. Participants also estimated the value of each stimulus, 4 times each, on a scale from 0 to 100.

The key feature of this task is that the mid-value option is compared more often to the lower-value option in the first context ($M_1$ versus $L_1$) and to the higher-value option in the second context ($M_2$ versus $H_2$). While the range adaptation and intrinsically enhanced models make similar predictions regarding test phase choice rates for L and H, they make different predictions for $M_1$ and $M_2$ choice rates. Regardless of which option the mid-range value is presented with, range adaptation models will rescale it based on the minimum and maximum value of the overall context's options (i.e., the average value of $L_1$ and $H_1$, respectively). Thus, they will show no preference for $M_1$ versus $M_2$ in the test phase. The intrinsically enhanced model, however, will learn to value $M_1$ more than $M_2$, as the former more often leads to goal completion (defined in our model as selecting the best outcome among the currently available ones; see Materials and methods) than the latter. This is because $M_1$ is more often presented with a worse option than $M_2$, which instead is more frequently presented with a better option. Thus, if participants follow a range adaptation rule (here indistinguishable from a classic RL model), there should be no difference between their choice rates for $M_1$, compared to $M_2$. By

contrast, if the intrinsically enhanced model better captures people's context-dependent learning of value options, $M_1$ should be selected more often than $M_2$ in the transfer phase.

We note that, while previous studies defined contexts as the set of available options at the time of a decision [12,13], here we adopt a more abstract definition of context, which comprises all the options that tend to be presented together—even when only 2 of them are available (a feature that was absent from previous task designs). Nonetheless, participants might have interpreted choice sets in which only 2 out of 3 outcomes were available as separate contexts. With the latter interpretation, range adaptation and intrinsically enhanced models would make the same predictions. To encourage participants to consider the 2 sets of stimuli as belonging to one context each (even when only 2 out of 3 options from a set were available), we designed a control study (M22B; $N = 50$, S2 Text) in which, instead of being completely removed from the screen, unavailable options were simply made unresponsive (S4 Fig). Therefore, participants could not select unavailable options, but the stimuli and outcomes associated with them were still visible.

**Ex ante simulations.** To confirm whether the intrinsically enhanced and the range$^z$ model could be qualitatively distinguished, we simulated participant behavior using the 2 models of interest with the same methods as described for data set B22. As expected, the intrinsically enhanced model showed higher choice rates for $M_1$ compared to $M_2$ (Fig 7A). By contrast, the range$^z$ model showed no preference for $M_1$, compared to $M_2$, in the test phase (Fig 7B).

**Behavioral results.** Overall, participants performed above chance in both the learning phase ($M = 0.90 \pm 0.02$, $t(49) = 16.08$, $p < 0.001$; S1 Fig) and the testing phase of the experiment ($M = 0.90 \pm 0.02$, $t(49) = 19.5$, $p < 0.001$).

Results matched all preregistered predictions. Despite the fact that options $M_1$ and $M_2$ were associated with the same objective mean value of 50, participants chose option $M_1$ (mean choice rate across all trials in the test phase: $0.57 \pm 0.02$) more often than option $M_2$ ($0.36 \pm 0.02$; $t(49) = 6.53$, $p < 0.001$ (Fig 8A and 8B). When the 2 options were directly pitted against each other, participants selected $M_1$ significantly more often than chance (i.e., 0.50;

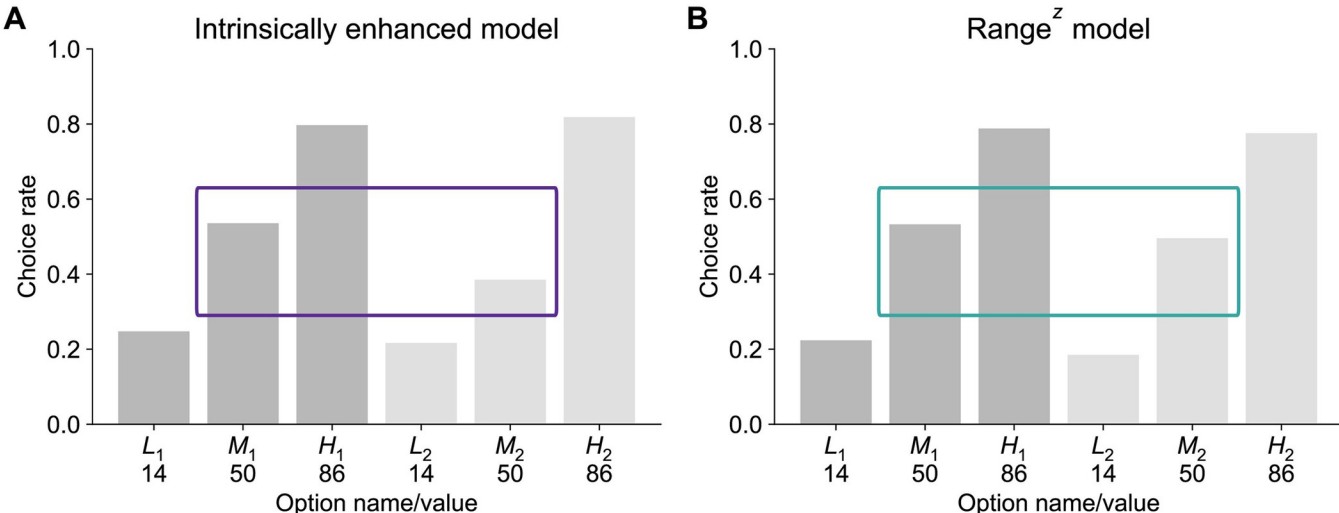

**Fig 7.** Ex ante model predictions based on simulations, for the intrinsically enhanced (**A**) and range$^z$ (**B**) models. Contexts 1 and 2 are shown in dark and light gray, respectively. The core prediction that differentiates intrinsically enhanced and range models is that participants will have a bias in favor of the middle option from context 1, compared to the middle option from context 2 (compare the purple and teal boxes). Simulation scripts used to produce this figure are available at https://osf.io/sfnc9/.

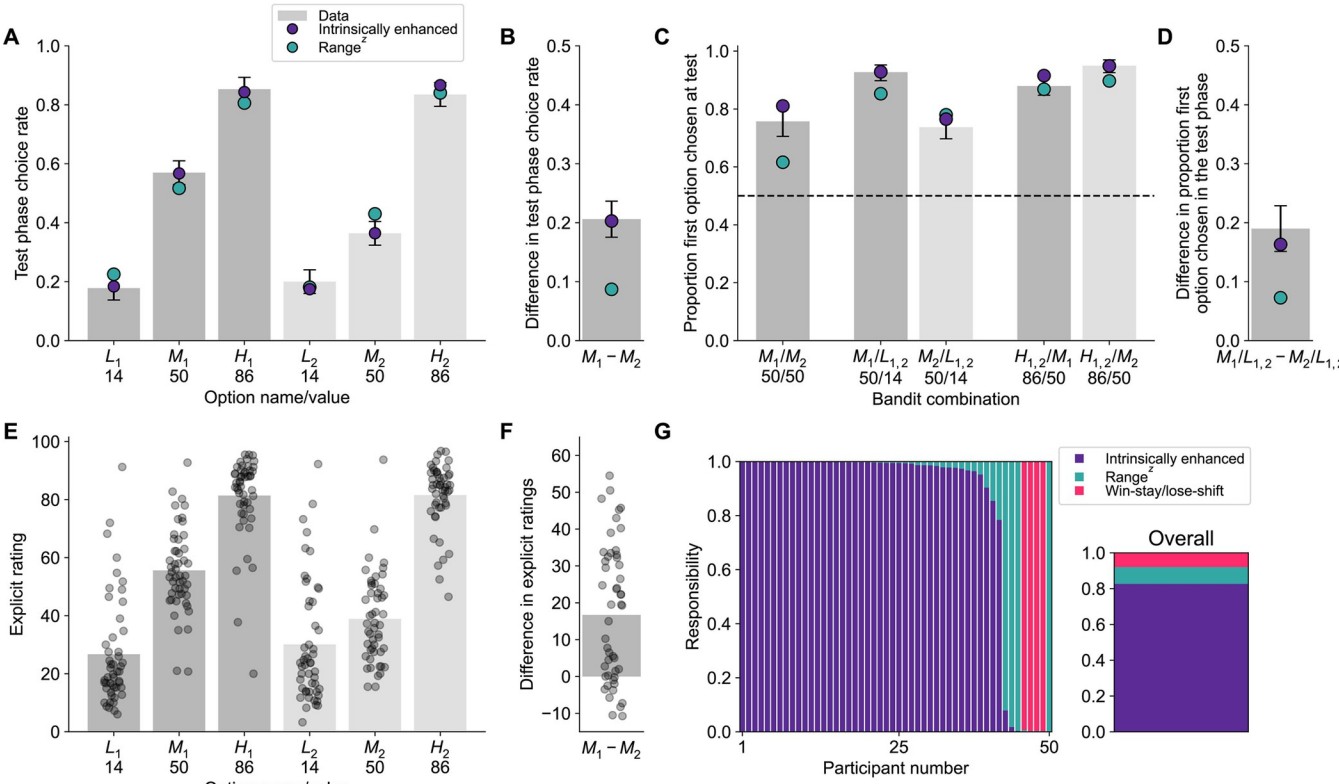

**Fig 8. Behavioral results and computational modeling support the intrinsically enhanced model. (A)** During the test phase, the mid-value option of context 1 (darker gray) was chosen more often than the mid-value option of context 2 (lighter gray), a pattern that was also evident in the intrinsically enhanced model's, but not the range$^z$ model's behavior. **(B)** Difference in test phase choice rates between stimulus $M_1$ and $M_2$. **(C)** When the 2 mid-value options were pitted against each other, participants preferred the one from context 1. When either was pitted against a low-value option, participants selected the mid-value option from context 1 more often than the mid-value option from context 2. When either was pitted against a high-value option, participants selected the high-value option from context 1 less often than the high-value option from context 2. The dotted line indicates chance level (0.5). **(D)** Difference between $M_1$ and $M_2$ in the proportion of times the option was chosen when compared to either $L_1$ or $L_2$. All these behavioral signatures were preregistered and predicted by the intrinsically enhanced, but not the range adaptation model. **(E)** Participants explicitly reported the mid-value option of context 1 as having a higher value than the mid-value option of context 2. **(F)** Differences in explicit ratings between option $M_1$ and $M_2$. **(G)** Model fitting favors the intrinsically enhanced model against range$^z$, as evidenced by higher responsibility across participants for the former compared to the latter. Data and analysis scripts underlying this figure are available at https://osf.io/sfnc9/.

M = 0.76 ± 0.05; t(49) = 5.13, $p < 0.001$; Fig 8C). Performance in the test phase was better for trials in which the $M_1$ option was pitted against either low option (M = 0.92 ± 0.03) than when $M_2$ was pitted against either low option (M = 0.74 ± 0.04; t(49) = 4.85, $p < 0.001$; Fig 8C and 8D). By contrast, performance in the test phase was better for trials in which the $M_2$ option was pitted against either high option (M = 0.88 ± 0.03) than when $M_1$ was pitted against either high option (M = 0.95 ± 0.02; t(49) = −4.07, $p < 0.001$; Fig 8C). Participants' explicit evaluations were higher for $M_1$ (55.65 ± 2.04) than $M_2$ (M = 38.9 ± 2.16; t(49) = 6.09, $p < 0.001$; Fig 8E and 8F).

Each of these behavioral signatures is expected from the intrinsically enhanced, but not the range adaptation model. Thus, the results of this study show direct support for the intrinsically enhanced model over the range adaptation model. These behavioral patterns were fully replicated in an independent sample (M22R; S2 and S3 Figs) and a control study where stimuli and outcomes for unavailable options were left visible (M22B; S5 and S6 Figs).

**Model comparison favors the intrinsically enhanced model.** In comparing competing models, we only considered the intrinsically enhanced and range$^z$ architectures. We also

included a basic strategy, namely the win-stay/lose-shift model, to capture variability that neither model could explain, as is considered best practice with HBI (see Materials and methods; [30]). The intrinsically enhanced model was the most frequently expressed (protected exceedance probability = 1) and had the highest responsibility across participants (intrinsically enhanced = 0.83, range adaptation = 0.09, win-stay/lose-shift = 0.08; Fig 8G). Even endowed with nonlinear rescaling of mid-value options, the range adaptation model failed to reproduce the key behavioral result that was observed in participants (Fig 8A–8D). We note that the range adaptation model showed a slight bias for the $M_1$ over the $M_2$ option, likely due to higher learning rates for chosen (compared to unchosen) outcomes and an experimental design where $M_1$ tends to be selected more often—thus acquiring value faster—than $M_2$. Nonetheless, the intrinsically enhanced model matched participants' behavior more closely than range adaption. Moreover, the intrinsically enhanced model's $\omega$ parameter (M = 0.46 ± 0.04) was significantly correlated with the difference in choice rates for $M_1$ versus $M_2$ in the test phase; Spearman's $\rho$ = 0.93, $p < 0.001$; S14D Fig. All modeling results were replicated in an independent sample (M22R; S1 Text and S2 Fig) and a control study in which stimuli and outcomes for unavailable options were not hidden from view (M22B; S2 Text and S5 Fig). Together, these results provide evidence for the higher explanatory power of the intrinsically enhanced model over range adaptation mechanisms.

## Discussion

In this study, we proposed an "intrinsically enhanced" model as a possible computational account of context-dependent reinforcement learning. There is now overwhelming evidence for the proposition that human value attributions are context dependent [5,32], and this general finding has been confirmed in RL tasks [7,11–13,17,33], wherein participants have to learn the value of each available option through trial and error, as is often the case in real-world scenarios. In a decision-learning problem, context-sensitive valuation results in value attributions that are relative to the alternative options available at the time of choosing. This phenomenon cannot be accounted for by a standard, unbiased RL architecture. Instead, context-sensitive valuation has been successfully captured by the range adaptation model, which normalizes absolute outcomes based on the minimum and the maximum value of options in the same context.

However, other models could explain the effects of context-sensitive valuation without resorting to range adaptation. Here, we explored one such alternative model whose premises are backed by extensive literature in broader psychological domains. Specifically, our account proposes that, after experiencing an outcome, people integrate its absolute value with an additional "all-or-nothing," internally generated signal of goal achievement (positive if the best available option was selected and 0 otherwise). Here, context-sensitivity is achieved by defining goal-dependent signals in relation to the best possible outcome. By analyzing 3 existing data sets totaling 935 participants and simulating choices from a fourth experimental design, we showed that the intrinsically enhanced model yields similar results to the range adaptation model, and should therefore be considered a valid alternative for explaining context-sensitive valuation in human reinforcement learning. Through an additional experiment intended to directly test differences between the competing 2 models, an independent replication of it, and a control study, we provided evidence for the stronger explanatory power of the intrinsically enhanced RL model over the range adaptation algorithm in some experimental settings. Specifically, participants selected more frequently an option that was more often the better one, even though it had the same objective value and occurred in a context of similar range.

The fact that the intrinsically enhanced model integrates both absolute and binary reward signals is consistent with the finding that, in the brain, outcome rescaling is partial, not

complete [34]. This poses the additional question of how the degree of outcome rescaling is set. While large inter-individual variability likely plays an important role in the extent to which extrinsic and intrinsic rewards impact value computation, task features may also affect how the contribution of each is balanced. For example, providing participants with complete feedback (as opposed to factual feedback only) induces an increase in relative outcome encoding [35]. In the intrinsically enhanced model, a free parameter ($\omega$) governs the extent to which absolute and goal-dependent values dominate in the computation of reward, with values of $\omega$ higher than 0.5 leading to stronger contributions of intrinsically generated, goal-dependent signals, and values of $\omega$ lower than 0.5 indicating a preponderance of absolute values in the calculation of reward. We have found preliminary evidence for the claim that the weighting parameter $\omega$ itself is dependent on task features in our analysis of the B21 [12] data set, which comprised 8 experiments. These differed in whether counterfactual outcomes were provided and whether contexts were presented in an interleaved or blocked manner. Both observing counterfactual outcomes and interacting with contexts in a blocked manner likely make it easier for participants to produce internally generated signals of goal achievement, as both features help them decide whether their selected option corresponded to the best available option in the current context. Thus, $\omega$ may be higher under these conditions than in experiments that followed a design with partial information or interleaved presentation of contexts. Indeed, we found that $\omega$ was higher in experiments with a blocked design than those with an interleaved presentation of stimuli. Moreover, $\omega$ was higher for experiments with complete feedback compared to those with partial feedback. These results provide initial evidence that differences in relative versus absolute outcome encoding based on experimental design could be accounted for by differences in emphasis on externally provided versus intrinsically generated reward signals.

Attentional biases and task demands could also affect the relative contribution of extrinsic and intrinsic reward signals in outcome valuation. For example, Hayes and Wedell [15] manipulated attentional focus in an RL task by asking participants to rate, at occasional time points in the task, either how they felt about particular options or the reward amount they expected to be associated with particular options. The latter condition resulted in blunted context-sensitive valuation during the transfer portion of the task, which may be formalized as a greater contribution of absolute reward values (i.e., the term participants' attention was brought towards) compared to internally rewarding features of the evaluated options (which, by contrast, was emphasized in the former condition). Similarly, Juechems and colleagues [16] found that, following a learning phase in which participants made within-context choices, exposure to decision sets that combined previously experienced contexts resulted in subsequent reductions in context-sensitive valuation. This behavior is thought to result from an adaptive response to task-contingent expectations and can be easily captured by an adjustment of the $\omega$ parameter once participants expect to make cross-context decisions.

Our findings corroborate a growing literature of theoretical advances [19–21] and experimental results [22–24,36,37] recognizing the fundamental role of goals in the computation of value in humans. In the classic RL literature, rewards have been understood as a property of the external environment that an agent interacting with it would simply receive [38]. While this approximation works well when modeling many behavioral tasks [39], it cannot explain why the context in which rewards are presented would matter to the computation of reward. Homeostatic reinforcement learning attempts to bridge this gap by proposing that the rewarding properties of an outcome are proportional to the extent to which it represents a discrepancy reduction from a physiological set-point [40]. However, the same principle could be extended to more cognitively defined target values. From crosswords to marathons, many human activities feel rewarding once completed despite not having immediate benefits on

survival [41]. One possible explanation for this phenomenon is that people actively set goals for themselves and, once these goals are active, progress towards them drives incremental dopamine release to guide the learning of actions that bring such goals about [42]. Indeed, reaching one's goal activates similar brain areas as secondary reinforcers, such as money or numeric points [22,43].

At face value, the need for multiple value signals may seem redundant. However, the idea that the brain computes multiple signals of reward is well established [44,45]. Goal-dependent signals in particular have been shown to coexist with neural signatures of absolute value encoding [23,24,36]. And, while it has received less formal recognition in cognitive psychology, the enhancement of standard RL frameworks with internally generated signals has led to notable breakthroughs in the field of artificial intelligence [46–48]. Classic RL architectures are provided with a hand-defined reward signal, which the agent seeks to maximize. However, learning solely based on these sparse rewards may be difficult. To circumvent this issue, artificial agents can be endowed with auxiliary reward functions to prompt the directed exploration of the environment based on intrinsic rewards. This approach has been shown to significantly improve performance even when extrinsic objectives are well defined (e.g., [47,49]). Inspired by work in developmental psychology, artificial systems have been developed that can even set their own goals and learn to reach them via self-defined reward functions, leading to self-supervised discovery and skill acquisition (see [50] for review). Thus, by integrating extrinsic rewards with self-generated, goal-dependent reward signals can critically enhance learning in both artificial and biological agents.

The intrinsically enhanced model of context-sensitive valuation has proven flexible enough to capture a host of behavioral findings, and powerful enough to produce novel hypotheses that were then confirmed by an ad hoc experiment. While we considered a variety of alternative algorithms to explain these phenomena (including the previously successful range adaptation and hybrid actor–critic architectures), the list of models we examined here is certainly not exhaustive. For example, divisive normalization—in which options are divided by their sum, rather than their range [3]—may represent a reasonable account of relative value encoding, although studies have shown it has even less explanatory power than range adaptation models [13]. Hayes and Wedell [51] recently proposed a frequency encoding model that computes outcome values based on their rank within the contextual distribution. Indeed, people's choices tend to be sensitive to the frequency with which an outcome is delivered, not just its average value [52]. In our model, the goal-dependent signal transmits the same amount of reward regardless of the absolute outcome value, such that the intrinsically enhanced model could also explain frequency-dependent biases through a much simpler heuristic. Future studies, however, may directly investigate similarities and differences between frequency-based and intrinsically enhanced RL models.

For ease of computation, in implementing the intrinsically enhanced and range adaptation models, we assumed that participants were always aware of the minimum and maximum value of available options, even for cases in which only the outcome of the chosen option was delivered, without counterfactual feedback. While [12] proposed a version of the range adaptation model that updates estimates for range extremes in a trial-by-trial fashion, we found that this extension did not improve behavioral fit and that the additional parameter was not recoverable (see S3 Text and S15 Fig). In addition to the lack of dynamicity, we make the simplifying assumptions that different aspects of outcome valuation are combined linearly and that goal-dependent signals are encoded as binary outcomes. It is not uncommon for researchers to assume that different aspects of an outcome are linearly combined during valuation [53,54]. Succinctly implementing goal-dependent outcomes as binary signals was not only sufficient to capture the data presented here, but also an approach often followed in artificial intelligence

research, which our theory was partly inspired by [50]. The brain is known to integrate various aspects of an outcome into a single value [55,56], and even seemingly incommensurable sources of rewards—such as food and monetary outcomes or social cues—are rescaled to similar values during cross-category comparisons [57]. Nonetheless, future modelers may consider expanding the intrinsically enhanced model with more accurate depictions of the internal processes allowing relative estimates of reward, and test specific assumptions we adopt here as a starting point. In addition, goal and reward signals were put on the same scale, capturing the assumption that the intrinsic reward for reaching a goal had equal value to the maximal extrinsic reward. A question for future neuroimaging experiments is how the brain may automatically compute such rescaling and escape commensurability issues. As our understanding of how goals are selected and goal-dependent rewards are adjusted to circumstances, intrinsically enhanced models of behavior could be improved with increased explanatory power. Along the same line of research, it may be possible to test for range adaptation as an adjunct mechanism to intrinsically enhanced reinforcement learning, as the 2 systems are not mutually exclusive.

A mathematically equivalent model to intrinsically enhanced reinforcement learning (in the form presented here) was proposed by [11], but not considered in later studies. There, the authors combined range adaptation with reference point centering, whereby an outcome is computed relative to the mean of all context-relevant outcomes. This mechanism provides a solution for punishment-based learning by bringing negative outcomes to a positive scale and then using them to reinforce behavior as in standard reinforcement learning [7]. The combination of range adaptation and reference point results in a binary signal that is numerically strikingly similar to, but theoretically distinct from, the internally generated reward signal of the intrinsically enhanced model. Neither intrinsic rewards nor a combination of range adaptation and reference point centering can, by themselves, explain behavior. As our model proposes, [11] combined relative outcomes with absolute ones in order to reproduce the same behavioral signatures displayed by human participants. Such a "hybrid" model thus encompasses multiple computational steps (range adaptation, reference point centering, and the mixing of relative outcomes with absolute ones). By contrast, the intrinsically enhanced model provides a more succinct explanation of how binary rewards can be computed. Crucially, the 2 models only overlap in specific instances, i.e., tasks in which only binary choices are presented and the same outcomes are associated, though with different probabilities, to options within the same context. This is not the case for the majority of tasks analyzed here.

In the formulation that we employed, the intrinsically enhanced model assumes that the participants' goal was to select the stimuli that yielded points most consistently or in larger amounts. This is clearly a simplification of the reality, in which participants likely had multiple, perhaps even conflicting goals (e.g., finishing the experiment as fast as possible, minimizing motor movements, choosing esthetically pleasing stimuli, and so on) that a complete account of participants' learning should account for [58]. In reality, participants' goals—even in simple tasks such as the ones described above—are also likely more nuanced than simple binary signals of whether the best available option was selected. These subtleties, while important, do not contradict our central message: that the prospect of achieving one's goals is key to the calculation of reward. Indeed, if it were possible to access the host of goals individual participants set for themselves when making a single task-related decision, absolute rewards may cover a less important role than previously thought in shaping behavior beyond their contribution to attaining goal-contingent outcomes. A major challenge for future research will be developing computational methods to infer goals and account for their contribution to the calculation of value.

In sum, we have illustrated how the context-dependent valuation of outcomes that is known to occur in human learners can be accounted for by an RL model that combines

externally delivered reinforcers with internally generated signals of goal attainment. Our re-analysis of 3 published data sets has provided evidence that such an intrinsically enhanced model can explain behavior similarly to range adaptation mechanisms, which have proven successful in the past. Moreover, by examining an additional experimental design, we show that the intrinsically enhanced RL model captures behavioral findings better than other competitors. Lastly, by qualitatively disentangling range adaptation and intrinsically enhanced mechanisms, we have shown evidence for the superiority of the latter in predicting and explaining context-dependent effects in human participants. Our findings point towards greater recognition of internal signals of rewards in formal theories of human reinforcement learning. By accounting for intrinsically generated rewards, extensions of the RL framework may lead to better models of context-dependence outcome valuation and beyond.

## Materials and methods

### Ethics statement

The experimental protocol was approved by the Institutional Review Board at the University of California, Berkeley (approval number 2016-01-8280) and conducted according to the principles expressed in the Declaration of Helsinki. For new data collection, formal, written consent was obtained from participants via an online form.

### Existing data sets

We used data and task structures originally collected and developed by [12] (B21), [11] (B18), [14] (G12), and [13] (B22). The key information about each data set is summarized in Tables 1 and S1. Readers interested in further details are referred to the original reports. Additionally, we designed a new experimental paradigm (M22) to address the distinction between intrinsically enhanced and range adaptation models more directly, the details of which are reported below.

### Original experiment (M22)

Our original experiment was specifically designed to differentiate between the intrinsically enhanced and range adaptation models. The study was preregistered, and a pilot experiment was conducted. Preregistration and pilot results are available at https://osf.io/2sczg/.

### Task design

The task design for M22 was inspired by [13] but adapted in order to distinguish between the range adaptation model and the intrinsically enhanced model. After reading the instructions and completing one or multiple training sessions (12 trials each) until they reached at least 60% accuracy, participants started the learning phase of the task, during which they were presented with a total of 6 stimuli belonging to 2 different contexts. Stimuli from the 2 contexts were presented in an interleaved fashion. On each trial, stimuli were presented either in pairs or in groups of 3, always from the same context, and participants were asked to indicate their preference by clicking on the chosen symbol. They then viewed the outcome of all available options, including the selected one—the latter surrounded by a square. Outcomes were drawn from a Gaussian distribution with a variance of 4 and a mean of 14, 50, or 86 (each mean being associated with a different stimulus in a given context). The specific values were chosen based on the following criteria: (1) matching [13] as closely as possible; (2) model simulations showing maximal differences in predictions between the models; and (3) piloting results. Options in the first context—stimuli $L_1$, $M_1$, and $H_1$ for low, medium, and high-value options of the

first context—were presented in one of the following combinations, 20 times each: $L_1$ versus $M_1$ versus $H_1$, $L_1$ versus $M_1$, and $L_1$ versus $H_1$. Options in the second context—stimuli $L_2$, $M_2$, and $H_2$—were presented in one of the following combinations, 20 times each: $L_2$ versus $M_2$ versus $H_2$, $M_2$ versus $H_2$, and $L_2$ versus $H_2$ (see Fig 6). Stimuli were randomly generated identi-cons as provided at https://github.com/sophiebavard/online_task. The total number of trials in the learning phase was thus 120. Whether each item was positioned on the left, right, or center of the screen was randomly determined on each trial. In the test phase, all possible pairs of options, including pairs of items across the 2 contexts and other combinations that were never shown before, were presented 4 times each, yielding a total of 60 trials. At the end of the experiment, participants also explicitly reported their estimates for the value of each stimulus, 4 times each for a total of 24 trials. Trial timing was identical to [13].

## Participants

Fifty participants (age M = 27.94 ± 0.75, age range = 18 to 40, 32% female) were recruited from the online platform Prolific (http://www.prolific.co), completed the experiment from their own devices, and were compensated $6.00 for their time (around 20 min, on average). Participants were incentivized to perform optimally in the task by informing them that the total number of points they obtained would be recorded. Following [13], no exclusion criteria were applied. A replication sample was recruited from the university's pool of participants (N = 55, as opposed to the preregistered aim of 50, due to overestimating the participants' drop-out rate; S1 Text). We also ran an additional control experiment that encouraged participants to view stimuli of the same type as belonging to the same context, even when one of the 3 options was unavailable (M22B; N = 50, S2 Text). Here, unavailable options were visible but unresponsive, rather than hidden (S4 Fig). The University of California, Berkeley's institutional review board approved the study.

## Behavioral analyses and hypotheses

We ensured that participants understood the task and learned option values appropriately by comparing the rate at which they selected the best option on each trial to chance levels using a single-sample $t$ test.

We compared test phase choice rates for the mid-value option of the first context of stimuli ($M_1$) to the mid-value option of the second context of stimuli ($M_2$) through a paired $t$ test. Following the predictions made by the intrinsically enhanced model, we expected that the $M_1$ option would be selected more often than the $M_2$ option.

In addition, we used $t$ tests to compare choice rates for test phase trials in which participants were presented with specific pairs of stimuli. In particular, we predicted that participants would select $M_1$ over $M_2$ on trials in which the 2 options were pitted against each other. Moreover, a secondary prediction was that correct choice rates would be higher for trials in which $M_1$ was pitted against $L_1$ or $L_2$ than trials in which $M_2$ was compared to $L_1$ or $L_2$. By contrast, the intrinsically enhanced model predicts that correct choice rates would be higher for trials in which $M_2$ was presented with $H_1$ or $H_2$ than those in which $M_1$ was compared to $H_1$ or $H_2$.

We additionally used paired $t$ tests to compare the average explicit ratings for $M_1$ versus $M_2$ in the last phase of the experiment. We predicted that participants would rate $M_1$ higher than $M_2$, following the intrinsically enhanced model.

## Models

**Base mechanisms.** All models shared the same basic architecture of a simple RL model [38]. Each model learned, through trial-by-trial updates, the EV ($Q$) of each chosen option ($c$)

within a set of stimuli (*s*), and over time learned to choose the option with higher *Q* most often. *Q*-value updates followed the delta rule [27], according to which the EV of the option chosen on a given trial (*t*) is updated as:

$$Q_{t+1}(s, c) = Q_t(s, c) + \alpha_c \cdot \delta_{c,t}, \tag{3}$$

where $\alpha_c$ is the learning rate for chosen options and $\delta_{c,t}$ is the reward prediction error for the chosen option on a given trial, calculated upon receipt of feedback (*r*) for that option as:

$$\delta_{c,t} = r_{c,t} - Q_t(s, c). \tag{4}$$

When complete (i.e., counterfactual) feedback was available, the EV of the unchosen option was also updated according to the same rule, but with a separate learning rate ($\alpha_u$) to account for potential differences in how feedback for chosen versus unchosen options is attended to and updated:

$$Q_{t+1}(s, u) = Q_t(s, u) + \alpha_u \cdot \delta_{u,t} \tag{5}$$

$$\delta_{u,t} = r_{u,t} - Q_t(s, u). \tag{6}$$

Given the Q-value of each option in a pair of stimuli, participants' choices were modeled according to the softmax function, such that the probability of selecting a stimulus on trial *t* ($P_t(s,c)$) was proportional to the value of the stimulus relative to the other options as given by:

$$P_t(s, c) = \frac{\exp(\beta \cdot Q_t(s, c))}{\sum_i \exp(\beta \cdot Q_t(s, c_i))}. \tag{7}$$

Here, $\beta$ represents the inverse temperature: the higher the value of $\beta$, the closer a participant's choice leaned towards the higher value option on each trial [26]. All models had separate $\beta$s for the 2 stages of the task ($\beta_{learn}$ and $\beta_{test}$).

*Q*-values were initialized at 0. All outcomes were divided by a single rescaling factor per experiment to be between [−1, 1] if negative outcomes were present, or between [0, 1] if outcomes were in the positive range, to allow comparison of fit parameters (in particular $\beta$) across models. For example, rewards were divided by 10 (for B21) or 100 (for B22 and M22). Note that this is simply a convenient reparameterization, not a theoretical claim—it is mathematically equivalent to a model where the outcome is not rescaled, the goal intrinsic value is equal to the maximum possible extrinsic reward, and $\beta$ is divided by the same scaling factor.

A model with a fixed learning rate of 1 and counterfactual learning was also implemented to capture the possibility that at least some of the participants were following a "win-stay/lose-shift" strategy, i.e., sticking to the same option after a win and shifting to the alternative after a loss [31].

Additional candidate models were the range adaptation model, the intrinsically enhanced model, and the hybrid actor–critic model. Range adaptation models capture the hypothesis that, when learning to attribute value to presented options, people are sensitive to the range of all available options in a given context. Intrinsically enhanced models capture the idea that participants may compute rewards not only based on the objective feedback they receive, but also on a binary, intrinsically generated signal that specified whether the goal of selecting the relatively better option was achieved. When the task design included loss avoidance, as well as gain trials (i.e., both negative and positive outcomes), intrinsically enhanced and range adaptation models had separate learning rates ($\alpha_{gains}$ and $\alpha_{loss\ avoidance}$) for the 2 conditions to account for potential differences between reward and punishment learning [59,60]. The hybrid actor–

critic model was included for completeness, as it was able to capture behavior best (including maladaptive test choices) in [14], but is not the main focus of our analysis. Next, we explain how each of these additional models was built. Additional variants of each model were initially tested, and later excluded due to their lesser ability to capture participants' behavior (S3 Text).

**Range adaptation models.** The range adaptation model [12] rescales rewards based on the range of available rewards in a given context (maximum, $r_{max}$, and minimum, $r_{min}$, values of the observed outcomes in a given set of stimuli, $s$) and uses these range-adapted rewards ($rr$) to update $Q$-values on each trial:

$$rr_t = \frac{r_t - r_{min}(s)}{r_{max}(s) - r_{min}(s)}. \tag{8}$$

Note that in [12], after being initialized at zero, $r_{max}$ and $r_{min}$ are updated on each trial in which $r$ is larger or smaller than the current value, respectively. The update of these range adaptation terms is regulated by a learning rate $\alpha_r$. In our modeling procedures, this additional parameter added unwarranted complexity and penalized the model. Therefore, we make the assumption that participants know the maximum and minimum outcome available in each context, based on the idea that these values are learned early on in the experiment. This assumption also prevents division by null values. In experiments with complete feedback, previous models have similarly assumed that participants update each context's minimum and maximum value at the first presentation of nonzero values [11]. For data sets in which up to 3 bandits could be presented in the same trial, an additional parameter, which we call $z$ ($\omega$ in [13]), was used to rescale mid-value options nonlinearly. This was done in accordance with the finding that, in behavioral experiments, participants' estimates (both implicit and explicit) of mid-value options were closer to the estimates for the lowest-value option than to those for the highest-value option [13]. This is achieved via range adaptation by simply elevating the $rr$ value to the power of the free parameter $z$:

$$rr_t = \left( \frac{r_t - r_{min}(s)}{r_{max}(s) - r_{min}(s)} \right)^z. \tag{9}$$

**Intrinsically enhanced model.** Intrinsically enhanced models combine the feedback from each trial with a binary, internally generated signal. Given the reward on the chosen ($r_{c,t}$) and unchosen options ($r_{u,t}$) on trial $t$, the intrinsic reward for the chosen option ($ir_{c,t}$) was computed as follows:

$$ir_{c,t} = \begin{cases} 0, & \text{if } r_{c,t} \neq r_{max,t}(s) \text{ or } (r_{u,t} \text{is known and } r_{c,t} < r_{u,t}) \\ 1, & \text{otherwise} \end{cases} \tag{10}$$

Thus, $ir_{c,t}$ was equal to 0 when $r_{c,t}$ was either different from the maximum available reward or less than $r_{u,t}$ (if the latter was known), and 1 when $r_{c,t}$ was either equal to $r_{max}$, or better than $r_{u,t}$ (if the latter was known). On trials with complete feedback, a counterfactual reward for the unchosen option ($ir_{u,t}$) was also computed according to the same principles:

$$ir_{u,t} = \begin{cases} 0, & \text{if } r_{u,t} \neq r_{max,t}(s) \text{ or } (r_{u,t} \text{ is known and } r_{c,t} > r_{u,t}). \\ 1, & \text{otherwise} \end{cases} \tag{11}$$

The terms $r$ and $ir$ were combined to define an intrinsically enhanced outcome ($ier$) used to update $Q$ values. A weight ($\omega$) determined the relative contribution of $r$s and $ir$s to $ier$s:

$$ier_{c,t} = \omega \cdot ir_{c,t} + (1 - \omega) \cdot r_{c,t} \tag{12}$$

such that Eq 4 and Eq 6 became, respectively:

$$\delta_{c,t} = ier_{c,t} - Q_t(s,c) \tag{13}$$

$$\delta_{u,t} = ier_{u,t} - Q_t(s,u). \tag{14}$$

For experiments in B21 in which participants received feedback during the test phase, the model was endowed with $\omega s$ for the 2 stages of the task ($\omega_{learn}$ and $\omega_{test}$), as we found that this improved the fit.

**Hybrid actor–critic model.** The hybrid actor–critic model combined a Q-learning module and an actor–critic module [14]. The Q-learning module was identical to the basic RL architecture described above, except for having the same $\beta$ parameter for both the learning and testing phase. This modification was intended to trim down model complexity. The hybrid actor–critic module comprised a "critic," which evaluates rewards within a given context, and an "actor," which chooses which action to select based on learned response weights. Prediction errors updated the critic's evaluations, as well as the actor's action weights. This allowed the actor to learn to select options without needing to represent their value explicitly. On each trial of the learning phase, the critic updated the value $V$ of a given context based on:

$$V_{t+1}(s) = V_t(s) + \alpha_{critic} \cdot \delta_{V,t} \tag{15}$$

$$\delta_{V,t} = o_{c,t} - V_t(s), \tag{16}$$

where $\alpha_{critic}$ is the critic's learning rate. The actor's weights for the chosen option $W$ were updated and then normalized (to avoid division by a null value) through the following rules:

$$W_{t+1}(s,c) = W_t(s,c) + \alpha_{actor} \cdot \delta_{V,t} \tag{17}$$

$$W_{t+1}(s,c) = \frac{W_t(s,c)}{\sum_i |W_t(s,c_i)|}. \tag{18}$$

As for the other models, actions were selected through a softmax. The values entered in the softmax function were a combination of the Q-learner's estimates and the actor's weights. A variable $h$ controlled the influence of Q-learner values on the actor's decisions, such that the higher the value of $h$, the higher the impact of Q-values on choices:

$$H_{t+1}(s,c_i) = (1-h) \cdot W_t(s,c_i) + h \cdot Q_t(s,c_i), \tag{19}$$

where H is the hybrid value upon which the actor's choices are based. In this model, $o_{c,t}$ was either the reward obtained on a given trial, or, for experiments where both gains and losses were possible, a function of the reward controlled by the additional loss parameter, $d$:

$$o_{c,t} = \begin{cases} 1-d, & \text{if } r_{c,t} > 0 \\ 0, & \text{if } r_{c,t} = 0 \\ -d, & \text{if } r_{c,t} < 0 \end{cases} \tag{20}$$

such that for $d = 0$ negative outcomes were completely neglected, for $d = 1$ positive outcomes were completely neglected, and for $d = 0.5$ the 2 types of outcomes were weighed equally. This transformation was performed for unchosen outcomes as well, whenever complete feedback was available.

## Ex ante simulations

For data sets B22 and M22, we performed ex ante (i.e., before obtaining any data) simulations of the expected behavior using the intrinsically enhanced and the range$^z$ models by drawing 100 sets of parameters from the following distributions. As in Bavard and Palminteri's [13] work, we sampled the $\beta$ parameter (inverse temperature) from a Gamma(1.2, 0.2) distribution and the $\alpha$ parameter (learning rate) from a Beta(1.1,1.1) distribution. The $z$ parameter for the range adaptation model was drawn from a Gamma(3, 3) distribution (which has a mean of 1, a lower bound at 0 and no upper bound), and the $\omega$ parameter (for the intrinsically enhanced model) was drawn from a Beta(2, 2) distribution (which has a mean of 1, a lower bound at 0, and an upper bound at 1).

## Model fitting and validation procedures

Models were fit via the HBI method as introduced by Piray and colleagues [30]. This statistical tool estimates parameters hierarchically by characterizing a population-level distribution from which individual values are drawn, while simultaneously comparing candidate models. HBI has been shown to provide more robust parameter estimates than more common methods (which are more prone to overfitting) and is less likely to favor overly simplistic models during model comparison [30].

All priors were set with a mean of 0 and a variance of 6.25. Inverse temperatures ($\beta$) and the range model's $z$ parameter were then transformed through an exponential function while all other parameters were transformed via a sigmoid function so as to be constrained between 0 and 1 (as is standard practice [30]). The HBI fitting pipeline involves 2 steps: first, the best-fitting parameters are estimated via Laplacian approximation (independent of the model space; [61]); second, Laplacian estimates are adjusted based on the likely contribution of each model for each participant's behavior. Parameters obtained via Laplacian approximation were used for the simulation step as HBI parameters are not always reliable for non-winning models [30]. Each subject's behavior was simulated 10 times with the same sequence of stimuli that was displayed to real participants. We report protected exceedance probability, which measures, for each proposed model, the probability that it is most commonly expressed in the studied population while ensuring that any difference in frequency among proposed models is statistically significant, and is thus a more conservative version of exceedance probability [62]. We also report each model's average responsibility, i.e., the probability that each model is responsible for generating each subject's data. In comparing candidate models, we also included at least one simpler strategy, compared to models of interest, to capture some of the variability, as this has been shown to improve HBI [30]. Data were fit to both the learning and test phases. In M22, we confirmed that fitting to only the learning phase did not change results and that phase-specific temperature parameters ($\beta$s) improved fit.

A prerequisite for drawing conclusions from the model fitting procedure is that at least the key models of interest are identifiable. To ensure that this was the case within our data sets, we computed and interpreted confusion matrices for each of them [28]. For each data set (excluding B22, since no data was available to us at the time of writing), this was achieved by retrieving the best-fit parameters for a given model, using them to simulate a new data set, and fitting all models on the simulated data set. The same procedure was then repeated for all proposed models. To ensure that simulation results were replicable, for data sets B18 and G12 ($N = 60$ and $N = 75$, respectively), double the original amount of participants were simulated; for M22 ($N = 50$), 4 times the amount of participants were simulated. For each fitting step, we report the model frequencies (i.e., the proportion of participants for which each model provided the best fit) and exceedance probabilities extracted from the HBI results. Across data sets B21,

B18, and G12, the true model was recovered more often than chance (i.e., 0.20 given that 5 models were compared simultaneously), with the lowest correct model recovery probability being 0.53 and all correct exceedance probabilities being 1 (S7A–S7C Fig). Model recovery was also satisfactory for the M22, M22R, and M22B data sets, with the true model being recovered more often than chance (i.e., 0.33) in all cases, the lowest correct model recovery probability being 0.55 and all correct exceedance probabilities being 1 (S7D Fig). The best-fitting parameters for the intrinsically enhanced model are reported in S8–S13 Figs.

## Supporting information

**S1 Table. Extended summary information for each of the data sets used for data analysis and/ or modeling.** Previously collected data sets were originally reported by [12] (B21), [11] (B18), [14] (G12), [13] (B22). For each data set experiment we used ("Exp."), we report: the rewards and probabilities associated with each context and bandit, as well as their expected value (EV); the key comparisons for which participants show irrational behavior, or for which models make specific predictions; whether feedback was partial or complete (i.e., including counterfactual) during learning; whether feedback was partial, complete, or absent during testing; whether bandit pairs were presented in a blocked or interleaved manner; whether there was a difference in the absolute magnitude of reward across bandit pairs ("Mag. Δ"); whether the task included negative outcomes ("Loss"); and the total number of participants in each original experiment ("N").
(PDF)

**S1 Fig. Learning phase performance in M22.** Participants learned to discriminate the correct option across bandit combinations in the learning phase. Data and analysis scripts underlying this figure are available at https://osf.io/sfnc9/.
(TIF)

**S2 Fig. Results from the independent replication of study M22 (M22R).** As in the main study (M22), behavioral results and computational modeling support the intrinsically enhanced model. **(A)** During the test phase, the mid-value option of context 1 (darker gray) was chosen more often than the mid-value option of context 2 (lighter gray), a pattern that was also evident in the intrinsically enhanced model's, but not the range$^z$ model's behavior. **(B)** Difference in test phase choice rates between stimulus $M_1$ and $M_2$. **(C)** When the 2 mid-value options were pitted against each other, participants preferred the one from context 1. When either was pitted against a low-value option, participants selected the mid-value option from context 1 more often than the mid-value option from context 2. When either was pitted against a high-value option, participants selected the high-value option from context 1 less often than the high-value option from context 2. The dotted line indicates chance level (0.5). All these behavioral signatures were captured by the intrinsically enhanced, but not the range adaptation model. **(D)** Difference between $M_1$ and $M_2$ in the proportion of times the option was chosen when compared to either $L_1$ or $L_2$. **(E)** Participants explicitly reported the mid-value option of context 1 as having a higher value than the mid-value option of context 2. **(F)** Differences in explicit ratings between option $M_1$ and $M_2$. **(G)** Model fitting favored the intrinsically enhanced model. Data and analysis scripts underlying this figure are available at https://osf.io/sfnc9/.
(TIF)

**S3 Fig. Learning phase performance in the M22 replication study (M22R).** Participants learned to discriminate the correct option across bandit combinations in the learning phase. Data and analysis scripts underlying this figure are available at https://osf.io/sfnc9/.
(TIF)

**S4 Fig. Task structure for experiment M22B.** In M22B, unavailable options were made more opaque and unselectable, but not completely invisible as was the case in M22 and M22R. The 2 tasks were otherwise identical.
(TIF)

**S5 Fig. Results from the control study for M22 (M22B).** As in the main study (M22) and the replication (M22R), behavioral results and computational modeling support the intrinsically enhanced model. See S2 Fig for caption details. Data and analysis scripts underlying this figure are available at https://osf.io/sfnc9/.
(TIF)

**S6 Fig. Learning phase performance in the M22 control study (M22B).** Participants learned to discriminate the correct option across bandit combinations in the learning phase. Data and analysis scripts underlying this figure are available at https://osf.io/sfnc9/.
(TIF)

**S7 Fig. Confusion matrices illustrating model recovery across data sets.** The upper row in each subplot shows model frequencies, the lower row shows protected exceedance probabilities. Model name abbreviations: IE = intrinsically enhanced, HAC = hybrid actor-critic, RL = unbiased, WSLS = win-stay/lose-shift. Data and analysis scripts underlying this figure are available at https://osf.io/sfnc9/.
(TIF)

**S8 Fig. Best fit parameters for the intrinsically enhanced model in B21.** Note that $\alpha_u$ could only be fit in experiments with counterfactual feedback, and $\omega_{test}$ could only be fit in experiments with counterfactual feedback at testing, and are therefore left at the initial prior. Abbreviations: the first letter in each triplet indicates whether feedback was partial (P) or complete (C) during learning; the second letter indicates whether feedback in the test phase was partial (P), complete (C), or not provided (N); the third letter indicates whether the experimental design was interleaved (I) or blocked (B). Error bars indicate the SEM. **(B)** Model responsibilities overall and across experimental conditions. Computational modeling scripts used to produce this figure are available at https://osf.io/sfnc9/.
(TIF)

**S9 Fig. Best fit parameters for the intrinsically enhanced model in B18.** Computational modeling scripts used to produce this figure are available at https://osf.io/sfnc9/.
(TIF)

**S10 Fig. Best fit parameters for the intrinsically enhanced model in G12.** Computational modeling scripts used to produce this figure are available at https://osf.io/sfnc9/.
(TIF)

**S11 Fig. Best fit parameters for the intrinsically enhanced model in M22.** Computational modeling scripts used to produce this figure are available at https://osf.io/sfnc9/.
(TIF)

**S12 Fig. Best fit parameters for the intrinsically enhanced model in the M22 replication study (M22R).** Computational modeling scripts used to produce this figure are available at https://osf.io/sfnc9/.
(TIF)

**S13 Fig. Best fit parameters for the intrinsically enhanced model in the control study for M22 (M22B).** Computational modeling scripts used to produce this figure are available at

https://osf.io/sfnc9/.
(TIF)

**S14 Fig. The $\omega$ parameter from the intrinsically enhanced model was significantly correlated with key behavioral signatures of context-sensitive learning. (A)** In B21, $\omega_{learn}$ correlates with the error rate in context 8 (choosing a bandit with EV = 0.75 vs. one with EV = 2.5. **(B)** In B18, $\omega$ correlates with the average error rate when choosing between a bandit with EV = 0.075 vs. one with EV = 0.25, and between a bandit with EV = −0.025 vs. one with EV = 0.025. **(C)** In G12, $\omega$ correlates with the average error rate when choosing between a bandit with EV = −0.1 vs. one with EV = 0.1, and between a bandit with EV = −0.2 vs. one with EV = 0.2. **(D–F)** In M22, M22R, and M22B, $\omega$ correlates with the difference in choice rates for bandits $M_1$ and $M_2$ (both had EV = 50). Spearman's $\rho$ is reported for each correlation. All behavioral biases were measured in the test phase. Computational modeling scripts used to produce this figure are available at https://osf.io/sfnc9/.
(TIF)

**S15 Fig. Adding an $\alpha_{range}$ parameter in the range adaptation model does not increase its responsibility.** Model validation **(A)** and comparison **(B)** by experimental condition in B21 with $\alpha_{range}$ in the range adaptation model. Here, the range adaptation model was endowed with an $\alpha_{range}$ parameter, which dynamically updated the minimum and maximum value of each context, as was done in the original study for data set B21 [12]. The addition of this parameter disadvantaged the range model (in teal) compared to the intrinsically enhanced model (in purple). See Fig 2 for caption details. Data underlying this figure are available at https://github.com/hrl-team/range/. Computational modeling scripts used for the illustrated results are available at https://osf.io/sfnc9/.
(TIF)

**S1 Text. Replication study (M22R) description.**
(PDF)

**S2 Text. Control study (M22B) description.**
(PDF)

**S3 Text. Preliminary model selection information.**
(PDF)

## Acknowledgments

We are indebted to Sophie Bavard, Stefano Palminteri, and colleagues for making their data publicly available. We are thankful to Amy Zou for help setting up the experiment online.

## Author Contributions

**Data curation:** Gaia Molinaro.

**Formal analysis:** Gaia Molinaro.

**Funding acquisition:** Anne G. E. Collins.

**Methodology:** Gaia Molinaro, Anne G. E. Collins.

**Resources:** Anne G. E. Collins.

**Supervision:** Anne G. E. Collins.

**Visualization:** Gaia Molinaro.

**Writing – original draft:** Gaia Molinaro.

**Writing – review & editing:** Gaia Molinaro, Anne G. E. Collins.

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
