## [Editor Report · Decision Letter 0]

16 Dec 2022

Dear Dr Molinaro, 

Thank you for submitting your manuscript entitled "Intrinsic rewards explain context-sensitive valuation in reinforcement learning" for consideration as a Research Article by PLOS Biology.

Your manuscript has now been evaluated by the PLOS Biology editorial staff and I am writing to let you know that we would like to send your submission out for external peer review. Please note that, due to the upcoming holidays, we have not been able to get feedback from either of the Academic Editors that we approached. We are therefore not yet making any formal call yet on whether the overall advance is appropriate for PLOS Biology or not and will be looking for reviewer support on this point. 

Before we can send your manuscript to reviewers, we need you to complete your submission by providing the metadata that is required for full assessment. To this end, please login to Editorial Manager where you will find the paper in the 'Submissions Needing Revisions' folder on your homepage. Please click 'Revise Submission' from the Action Links and complete all additional questions in the submission questionnaire. Please note that I will be on holidays starting on Monday so we also will not start lining up reviewers on this work until the first week of Jan 2023.

Once your full submission is complete, your paper will undergo a series of checks in preparation for peer review. To provide the metadata for your submission, please Login to Editorial Manager (https://www.editorialmanager.com/pbiology) within two working days, i.e. by Dec 21 2022 11:59PM.

Kind regards,

Kris

Kris Dickson, Ph.D., (she/her)

Neurosciences Senior Editor/Section Manager

PLOS Biology

kdickson@plos.org

---

## [Decision Letter · Decision Letter 1]

8 Feb 2023

Dear Dr Molinaro,

Thank you for your patience while your manuscript "Intrinsic rewards explain context-sensitive valuation in reinforcement learning" was peer-reviewed at PLOS Biology. It has now been evaluated by the PLOS Biology editors, an Academic Editor with relevant expertise, and by several independent reviewers. 

In light of the reviews, which you will find at the end of this email, we would like to invite you to revise the work to thoroughly address the reviewers' reports. Please note that, given the extent of revision needed, we cannot make a decision about publication until we have seen the revised manuscript and your response to the reviewers' comments. Your revised manuscript is likely to be sent for further evaluation by all or a subset of the reviewers.

**IMPORTANT - SUBMITTING YOUR REVISION**

*Re-submission Checklist*

*Published Peer Review*

*PLOS Data Policy*

*Blot and Gel Data Policy*

Sincerely,

Kris

Kris Dickson, Ph.D., (she/her)

Neurosciences Senior Editor/Section Manager

PLOS Biology

kdickson@plos.org

REVIEWS:

Do you want your identity to be public for this peer review?

Reviewer #1: No

Reviewer #2: No

Reviewer #3: Yes: Sophie Bavard

Reviewer #1: This manuscript from Molinaro and Collins proposes and tests a new computational model for context-sensitive value learning. While context-dependent valuation has largely been examined during decision making (with known option values), recent work has shown that context sensitivity also arises during trial-and-error based reinforcement learning (RL), with much of the literature (Palminteri and others) proposing range adaptation (RA) as a computational explanation. Here, the authors propose an alternative intrinsically enhanced RL model (IEM). in which object valuations are a mixture of classic outcome-based value and an intrinsic reward signal based on goal achievement. Across both previously published datasets and new data, the authors find that the IEM model explains context-sensitive valuation as well as or better than RA.

Context-dependent valuation is an important topic in fields like psychology, economics, and computational neuroscience: in contrast to standard normative theories assuming an absolute valuation process, humans show valuation that is context-sensitive and varies with the other current or previously available options. I quite like this paper and its overall approach, proposing a novel, psychologically-motivated, computationally-specified model for reinforcement learning that offers an alternative to range adaptation. The HBI approach to model fitting and comparison seems sophisticated and theoretically well justified, though I do have some questions about how final results (e.g. group parameter estimates) rely on the space of models compared. In addition, i think the authors could clarify a few issues about the form of their model and fit mixture parameter values. 

Major points

(1) Justification for the specific mixture model formulation. The authors do a good job of discussing the general support/evidence for goal-sensitive valuation, but could provide more justification for the specific form of the model they propose. While the idea of a simple mixture model is intuitive, there are some important issues that should be at least addressed. For one, is there a conceptual or empirical reason to expect that values are derived as a (linear) mixture of classic RL-derived estimates and an intrinsic goal signal? Second, is there a particular reason why the intrinsic goal signal is construed as a binary all or nothing signal?

One thing that feels missing from the paper is a discussion of what the intrinsic reward signal provides that allows it to capture similar behavior explained by range adaptation: since the IEM model mixes absolute reward and intrinsic reward signals, presumably the intrinsic reward is generating the context-sensitivity. Does this result from the nature of the tasks used here (with learning and test phases), or does this reflect a general relationship between goal attainment and contextual scaling?

(2) Consistency between replicating experimental data and model fitting metrics. There are several instances where the authors state that the IEM model better fits the behavioral data but the model comparison favors the RA model, for example the B18 dataset (IEM more closely matches data but RA is the most frequently expressed model) and the M22 replication dataset (text_s1). Can the authors clarify why there is this discrepancy where the group level parameters seem to favor the IEM even though across individuals the IEM is not the winning model?

(3) Presenting model parameters. The crucial parameter in the IEM model is the mixture parameter omega, and I think it would be informative if the authors presented best fit model parameters (either group level, or average population) across the different datasets. The authors briefly discuss how omega varies across different task variants in the B21 dataset (section 1.3.4), but it would be useful to see how omega varies across datasets. I realize that some of this information is plotted in the SI (e.g. Fig S5), but given its central role in the model I'd like to see information about omega in the main text.

Related points: Have the authors examined whether the variability in omega (as seen in fFg s5) correlates with behavioral measures of context-dependence (or other general behavioral performance features) across individuals? And is there a specific reason that in certain no feedback test phases (PNI, CNI, etc.) the omegas cluster at 0.33? I presume this was set up as such, but it would be good to see it stated.

(4) Clarification of model fitting and simulation results. As I understand it, the authors used hierarchical Bayesian inference (HBI) to fit the different models, which estimates model parameters with population distributions while simultaneously accounting for the assignment of different models to different individuals. First, can the authors clarify what they mean when they simulate data from different models (e.g. Fig. 2A, Fig 3 left panel, Fig. 4 left panel): are these the result of group level model parameters? And do these group level parameter estimates for a given model include an effect of all individuals, even though some (or many) of those individuals have other models with the highest responsibility? I believe that HBI weights the group level estimates accordingly, but it would be good (A) for this information to be specified in the Methods or Results somewhere, and (B) for the figure legends to be more clear about what is being plotted. Second, at a more conceptual level, does this approach mean that group level parameter estimates will depend at least partially on what companion models are included in the fitting process? 

Minor points

(1) Typo, line 766: "when it r_c,t"

(2) In the Methods description of the hybrid actor-critic model,it's not entirely clear how the parameter h controls the integration of Q-learner and actor-critic values. Perhaps stating the equation would be helpful. 

(3) For the analysis of the B22 dataset (Figure 5), ti would be helpful if - in addition to the simulated predictions - the authors could show in the figure the empirical behavioral results.

(4) For the ex-ante simulations for B22 and M22 (1.11), the authors draw model parameters from distributions. It would be helpful if the authors stated in teh text a brief rationale for those distributional assumptions.

(5) Dissociating the IEM model from range adaptation and reference point centering model. As the authors discuss, the hybrid range adaptation model with reference point centering (from Bavard et al 2018) is mathematically very similar to the IEM, in that they implement a mixture model of absolute and relative reward signals. I actually think the authors could state the advantage of their model a little more strongly: the binary signal used in the Bavard paper is a similar binary signal, but only in specific scenarios where stimuli are offering the same reward with different probabilities of reward (or nothing). Thus, the concept of an intrinsic goal signal is much more general that the very specific from of binary relative value used in Bavard et al.

(6) For Figure 2, it would really help if the figure had a legend explaining the different letter designations (PNI vs PPI vs CNI). I realize this is in the figure legend, but it would help if it were in the figure image itself.

------------------

Reviewer #2: In this manuscript, the authors proposed a new reinforcement learning model enhanced by an intrinsic reward component. Reinforcement learning is a highly influential framework. Although its core computational principles are consensual, how reward is represented is still debated. Recent developments in the fields include range adaptation, a phenomenon consisting in rescaling the value function to remain sensitive to the current range of rewards that available in the environment. That is, action outcome would not be reinforcing per se; instead, a normalised outcome (by the reward range) would reinforce the action. As a consequence, people tend to be reinforced to the same extent by the best possible option in a given context, regardless their absolute expected values. Here, the authors proposed a new computational model enhanced by an intrinsic reward component: the outcome value would depend on the agent's goal achievement. The model specifically assumes that participants' goal is to have the best possible performance in a given trial and therefore to choose the best possible option. That is, an outcome reinforces the action to the extent that it corresponds to the best possible outcome available to the participants. 

I really enjoyed reading this manuscript for many reasons. I found it to be clearly written. I found the new computational model exciting, as well as the idea to equip the model with an intrinsic reward component. I liked the simulations, the model fit procedure and the model selection metrics (with the use of estimated model frequencies and protected exceedance probabilities) as well as the model recovery analysis. Also, the original experiment has been preregistered.

I however have some concerns which may weaken the results and some suggestions which I hope will help improve the manuscript. 

My first concern is related to the results from the authors' original experiment, the only one in the manuscript that can really discriminate between the range adaptation model and the intrinsic reward enhanced model. 

1) In the framework of value-based range adaptation, a context is defined to my understanding by the cues that are presented together and is therefore called a "choice context" (see e.g. Palminteri et al., 2015). For example, if a, b and c are always presented together as well as d, e and f, then two contexts are defined. In the authors' original experiment and as per this definition, each trinary (e.g. L1-M1-H1) and each binary choice (e.g. L1-M1) represents a choice context. I guess that's why when three options are used to define a context, they are always presented together in Bavard et al., 2022. A more abstract definition (that the authors seem to embrace) would correspond to the abstract set of stimuli that can be presented together (e.g.L1, M1, H1) even when only two cues of that set are displayed (e.g. M1-H1). I guess that a block design (i.e. L1-M1-H1, M1-H1, L1-H1) would create enough temporal contiguity to create a context corresponding to the latter definition, as would the use of a contextual cue. Yet, in an interleaved design (as in this new experiment), it seems to me that the former definition would apply more (choice context). This is not a purely semantic or conceptual debate, because it has a crucial consequence regarding the range adaptation model in this experiment. The key prediction relates to two pairs of cues -L1-M1 in context 1, M2-H2 in context 2. In the transfer phase and according to the intrinsic reward enhanced model, participants should choose more often M1 over M2 because M1 is the best possible option in the given set of option. Yet, range adaptation could (should) occur within a choice context. That is, it would occur for a trinary trial (e.g. L1-M1-H1), as well as within a binary trial (e.g., L1-M1, M2-H2). In such a case, the range adaptation model leads to the exact same prediction as the intrinsic reward enhanced model, weakening considerably (if not cancelling) the evidence supporting the latter. Indeed, M1 would be reinforcing when presented with L1, unlike M2 presented with H2, and therefore M1 would be chosen more often in the transfer phase than M2, as the authors found. 

Perhaps a cleaner design would be to always present all three options on the screen, defining a clear context 1 & 2. In some trials, participants could only choose between two options (although all three would be displayed) to implement the crucial manipulation from the original experiment. The range adaptation model predicts that the middle options would always be small, while the intrinsic reward enhanced model would predict that the M1 is more selected than M2. 

2) Given that the learning rate for the chosen option is usually higher than the learning rate for the unchosen option, I wonder whether the unbiased model would predict the same pattern of results as the intrinsic reward enhanced model: M1 would be updated with a higher learning rate than the unchosen M2 in the M2-H2 pair, leading M1 to have a higher value than M2. I don't think this model is included in any of the reported model space. 

3) Another major concern I have is related to the comparison between the range adaptation model and the intrinsic reward model. The latter is actually a mixed of the basic model (where the reward is the reward magnitude) and of the intrinsic reward component. A fair comparison would include a mixed of the basic model and of the range adaptation model, as in Bavard et al.,2018 (see equation 2). Alternatively, I suggest to compare a pure intrinsic reward model with the range adaptation model. In this way, the model selection would be much fairer to distinguish between the range adaptation model and of the intrinsic reward model.

4) I also suggest to avoid to include too many models in a first model space, to avoid a potential spread of the models in the population that would otherwise be explained by e.g. the range adaptation model. Therefore, I suggest to compare first the range adaptation model with a pure intrinsic reward model, then the mixed range adaptation model and the mixed intrinsic reward model. That would be 2 model spaces with 2 models. Additionally, another model space could include those 4 models and a family model comparison may be performed. Only then a broader model space would be more informative. 

Reviewer #3: In this study, Molinaro & Collins investigate the computational form of context-dependent reinforcement learning in humans. Over the analysis of previously published datasets (from 2 different research groups), ex-ante simulations on unpublished data, and the analysis of a new dataset using a similar - yet novel - reinforcement learning task, the authors compare different candidate models implementing context-dependent learning in different forms (mainly, intrinsically enhanced and range adaptation models). Model fitting, comparison and falsification slightly favored the intrinsically enhanced model over other candidates in most of the datasets. Overall, this represents interesting work, clear and well written. Strengths of the paper include the re-analysis of several datasets coupled with ex-ante simulations AND a novel task, as well as several candidate normalization models. However, I have several major concerns about model implementation and interpretation that, I believe, limit the study's contribution to the literature, in its current state.

1. Intrinsically enhanced model. The model is defined as a weighted sum between the absolute experienced outcome (which is not really true - see my next point) and a relative measure (whether the outcome is maximum within its own context). Mathematically, the model is equivalent to the already proposed "hybrid" model in B18 to account for partial adaptation - and the results reported here are therefore similar for this dataset (including the analyses on the weight parameter, see B18 Fig. 4). Yet, this similarity is only mentioned in the discussion (line 574) and mostly overlooked over the whole paper. This implementation had some major issues that lead us to move to range adaptation: the weighted sum is a descriptive model by nature and not very plausible on a cognitive level. Previous studies with similar paradigms rather support the hypothesis that contextual information is integrated in a single brain region, as partial (or full) adaptation rather than dual valuation (Padoa-Schioppa, 2009; Palminteri et al. 2015; Rigoli et al. 2016; Burke et al. 2016). This evidence, coupled with the dynamic propensity of adaptation over the task time (see B18, Fig. 4a), placed range adaptation (as described in B21) as a good candidate model: participants first start encoding the absolute outcome, and as they progressively learn context values, are able to recognize what constitutes the best choice in each context. Here, a lot of assumptions are made implying that participants already know the structure of the task (the context/maximum values are known from the beginning, not learned over trials), which might not be the case (see B18 Fig. 4e-f).

2. Commensurability issue. One other arguable feature of the intrinsically enhanced model is that the outcomes have to be rescaled in order for it to make sense. As stated in the methods (line 714), the outcomes are rescaled between -1 and 1 to allow comparison of fit (temperature) parameters. I however disagree with the next sentence (line 717-718): this could be a theoretical claim in the sense that if the values are not rescaled in a task such as B22 or M22 where the outcomes have a much higher magnitude, then the 'absolute' component of the model would overweight the 'relative' one, the latter being always bounded between 0 and 1. Even by assuming that the participants somehow know the outcome boundaries from the beginning of the task, one of the components would have to be rescaled, which is a drawback not implemented in the range adaptation model. 

2B. To make the two models comparable, one possibility could be to fit the intrinsically motivated model without assuming the normalization of the absolute outcomes. It could still work by adjusting the weight parameters, but then the authors will have to explain why (inevitably) the weight parameters change radically from an experiment to another on (as a function of the actual magnitude of the rewards at stake; a problem that does not arise with the range adaptation model).

2C. I also think it would be important to present in the main text the model comparison results showing that the RANGE model better fits the majority of participants in the replication experiment (Figure S2). 

3. Range adaptation model predictions. Given the implementation of the range adaptation model in this paper, the model should be able to reverse participants' ability to select the highest EV stimulus in test trials with ΔEV = 1.75 in the experiments where complete feedback is provided during the test phase (B21), but the model simulations suggest otherwise (see Figure 2A). This does not square with the results reported in B21 and rather suggests a potential error in the fitting or the simulations that may not take the feedback provided the test phase into account.

4. Range adaptation implementation. The range adaptation model implemented in the present work differs from that originally presented in B21, in that it assumes that the actual range is perfectly known from the beginning and not learned with a dedicated learning rate (alphaRANGE). The authors mention that this parameter was not included to avoid over penalization and because it was not very well recoverable. However, the fact that the exact value of the parameter is not - or weakly - recoverable does not necessary imply that the additional degree of freedom/flexibility is not recoverable or justified that model space level. In other terms, weak parameter recovery for alphaRANGE could still be associated with good model recovery for a model using the alphaRANGE. This could be the case when, for example, the choice patterns generated by models with alphaRANGE=0.2 and alphaRANGE=0.4 are very similar (which is probably the case: Rmax and Rmin should converge quickly), BUT still quite different from those of a model with alphaRANGE=0.0. Even though I have no reason not to trust the authors concerning the fact that the addition of this parameter penalized the model and made no radically difference, since the goal of the paper is comparative, it would be informative to include this parameter, at least in the Supplementary materials and at least for the where it was originally proposed datasets (B18, B21). This is also important because currently the comparison between the range adaptation model and the intrinsically enhanced model confounds two factors: (1) the architecture (which is what the authors are interesting in) and (2) the fact that the normalization process is assumed to flexible in one case (thanks to the weight parameters) and inflexible in the other case (range adaptation model). 

5. Context definition in the novel task. In M22, the reader will benefit from a more precise definition of what constitutes a context. In previous studies, the context would be composed of 2 or 3 alternatives, stable over the learning part of the task. Here, different choice problems (choosing between options A/B/C, or A/B, or A/C) are treated as one context. My understanding is that the contexts values are the ones of the trinary choice, but this might not be the case (see e.g., the debriefing questions in B18 Fig. 4e-f, where some participants did not even notice that the options were arranged in fixed pairs). The authors could consider the possibility that if the context is defined by the options it includes, then if the 'trinary maximum' is not there, the context changes and the 'trinary second best' becomes maximum. In such a setting, even a simpler form of range adaptation (such as the one proposed in B22) could make similar predictions as the intrinsically enhanced model.

Minor points

6. line 712. Is there any reason why the authors chose to fit different temperature parameters for both phases of the task? This suggests that the optimization was performed on the whole dataset, which was not the case of B18 and B21 where it was done on the learning phase data only. Would the intrinsically enhanced model perform equally well in the test phase if this was the case?

7. line 733. Separating learning rates between conditions also makes the assumption that the participants have a clear understanding of the task structure. Alternative implementations include separating learning rates with the sign of the prediction error, which should not produce a huge difference in the results.

8. line 751. The division by null values was avoided in previous studies by adding 1 to the denominator, which also allowed the models to be nested within each other.

9. Hayes and Wedell (2022) recently showed that context effects can be "top-down" modulated by focusing the participants attentions on internal or external features; Juechems et al. (2022) recently showed that context effect can be modulated "bottom up" by implicitly instructing subjects of the benefit of absolute encoding. It would be nice if the authors discuss these results in the light of the intrinsically motivated model. 

Signed,

Sophie Bavard

---

## [Editor Report · Decision Letter 2]

28 Apr 2023

Dear Dr Molinaro,

Thank you for your patience while we considered your revised manuscript "Intrinsic rewards explain context-sensitive valuation in reinforcement learning" for publication as a Research Article at PLOS Biology. This revised version of your manuscript has been evaluated by the PLOS Biology editors and by the Academic Editor who is fully satisfied by the revision and appreciates your comprehensive response to the reviewers. 

Based on our Academic Editor's assessment of your revision, we are likely to accept this manuscript for publication. However, before we can editorially accept your manuscript, we need you to address the following data and other policy-related requests.

EDITORIAL REQUESTS: 

1) FINANCIAL DISCLOSURES: I noticed in your financial disclosure that you report having received no specific funding for this work. Even if you did not have a grant specifically written to fund this project, we would require that you acknowledge the sources of money that funded this work, including salaries, etc. You should also acknowledge institutional funding or private grants as well, if relevant.

2) ETHICS STATEMENT: Please update your ethics statement, in the methods section, to include an approval number. Please also indicate whether the experiments were conducted according to the principles expressed in the Declaration of Helsinki.

3) DATA AVAILABILITY: Thank you for depositing your data on OSF - for some reason I could not access this data to check that it meets our requirements for data sharing. Can you please provide me a reviewer token, or accept my request to view the data? **Please also ensure that figure legends in your manuscript include information on where the underlying data can be found (by referencing the relevant OSF links).

For more information on our data availability requirements, see here: http://journals.plos.org/plosbiology/s/data-availability. Also see this editorial: http://dx.doi.org/10.1371/journal.pbio.1001797

We expect to receive your revised manuscript within two weeks. 

*Published Peer Review History*

*Press*

Sincerely,

Luke

Lucas Smith, Ph.D.

Associate Editor,

lsmith@plos.org,

PLOS Biology

---

## [Editor Report · Decision Letter 3]

23 May 2023

Dear Dr Molinaro,

Thank you for your patience while we considered your revised manuscript "Intrinsic rewards explain context-sensitive valuation in reinforcement learning" for publication as a Research Article at PLOS Biology and thank you also for addressing our previous editorial requests in this most recent revision. 

As discussed, over email, before we can accept your study, we would like to invite you to revise the manuscript further, to address a comment from Reviewer 3, who flagged a potential issue after we invited the last revision. Reviewer 3's comments are appended below. 

We understand from our correspondence over email, that you were able to address this issue without and substantial changes to the conclusions of the study - however before we assess the changes made in detail, we think it would be helpful to have you update your manuscript with the relevant changes. 

We expect to receive your revised manuscript within two weeks. Please reach out if you have any questions. 

*Published Peer Review History*

*Press*

Sincerely,

Lucas

Lucas Smith, Ph.D.

Associate Editor,

lsmith@plos.org,

PLOS Biology

REVIEWER COMMENTS: 

Reviewer 3:

Thank you for letting me know and for sharing the rebuttal letter. I have taken the time to read it and I appreciate the authors' response to my concerns. However, I would have liked to have an opportunity to engage further with the authors regarding some analytical choices that I think could still be improved.

I would like to bring to the academic editors' attention a potential issue with the simulations presented in the rebuttal letter. Specifically, the authors' response to my 4th point is much likely to be invalid, as the simulations seem to suffer from the same code bug found in the previous ones (which was correctly identified and highlighted in my 3rd point). This claim can be easily checked by noting how the presented simulations of the RANGEalpha model (which is supposed to be exactly the same as our model published in Science Advances in 2021) differ from our own published simulations, despite declaring very similar methods (Figure S15 in the rebuttal letter). I believe such a potential mistake should be carefully examined to ensure the accuracy of the comparison between the two models.'

---

## [Decision Letter · Decision Letter 4]

15 Jun 2023

Dear Dr Molinaro,

Thank you for the submission of your revised Research Article "Intrinsic rewards explain context-sensitive valuation in reinforcement learning" for publication in PLOS Biology and for addressing the last issue identified by Reviewer 3 in this most recent revision. Your revised manuscript has been assessed by the PLOS Biology editorial team, the Academic Editor, and by reviewer 3, and we are all satisfied by the changes made. Reviewer 3's comments are appended below my signature. 

Therefore, on behalf of my colleagues and the Academic Editor, Christopher Summerfield, I am pleased to say that we can in principle accept your manuscript for publication, provided you address any remaining formatting and reporting issues. These will be detailed in an email you should receive within 2-3 business days from our colleagues in the journal operations team; no action is required from you until then. Please note that we will not be able to formally accept your manuscript and schedule it for publication until you have completed any requested changes.

PRESS

Sincerely, 

Lucas Smith, Ph.D.

Senior Editor

PLOS Biology

lsmith@plos.org

REVIEWER COMMENTS

Reviewer 3, Sophie Bavard (note: this reviewer has signed their review): I would like to thank the editorial team for forwarding my concern, I do not have any further comments. We thank the authors for their response and their interest in this research question and look forward to follow up on this discussion.